# On ranking *via* sorting by estimated expected utility

**Clément Calauzènes**
Criteo AI Lab
Paris, France
c.calauzenes@criteo.com

**Nicolas Usunier**
Facebook AI Research
Paris, France
usunier@fb.com

## Abstract

Ranking tasks are defined through losses that measure trade-offs between different desiderata such as the relevance and the diversity of the items at the top of the list. This paper addresses the question of which of these tasks are asymptotically solved by sorting by decreasing order of expected utility, for some suitable notion of utility, or, equivalently, *when is square loss regression consistent for ranking* via *score-and-sort?* We answer to this question by finding a characterization of ranking losses for which a suitable regression is consistent. This characterization has two strong corollaries. First, whenever there exists a consistent approach based on convex risk minimization, there also is a consistent approach based on regression. Second, when regression is not consistent, there are data distributions for which consistent surrogate approaches necessarily have non-trivial local minima, and for which optimal scoring function are necessarily discontinuous, even when the underlying data distribution is regular. In addition to providing a better understanding of surrogate approaches for ranking, these results illustrate the intrinsic difficulty of solving general ranking problems with the score-and-sort approach.

## 1   Introduction

The usual approach in learning to rank is to score each item (e.g., a document) given the input (e.g., a search query), and produce the ranking by sorting in decreasing order of scores. This score-and-sort approach follows the probability ranking principle of information retrieval [29], which stipulates that documents should be rank-ordered according to their estimated probability of relevance to the query.

In practice, the definition of a "good" ranking requires more than estimates of relevance. For instance, in scenarios where several users issue the same query "jaguar" but with different intended meanings (e.g., the animal or the car brand), it is desirable to produce diverse rankings where each user finds a relevant document as early as possible. While in the probability ranking principle, items are ranked in decreasing order of their *expected utility* to the users, in sophisticated ranking tasks with a trade-off between relevance and diversity, the definition of a utility per item is not trivial, if not impossible.

In this paper, we study what ranking tasks are solved via sorting by expected utilities, in a general supervised ranking framework that captures different types of ground-truth signal and losses. Since utilities can serve as target values to learn the scoring function through square loss regression, the optimality of sorting by expected utilities is equivalent to the consistency of regression. The main question we address is thus: *When is square loss regression consistent for ranking* via *score-and-sort?*

The consistency of regression for ranking, and more generally the consistency of convex risk minimization for ranking, are still only partly understood. Existing consistency results only apply to gain-based losses such as the Discounted Cumulative Gain and precision and recall at $K$ [12, 28, 6, 19, 14], for which there is an explicit utility function [see e.g., 8, Table 1]. For other losses, only impossibility results are known. Duchi et al. [15], followed by Calauzènes et al. [7] and Ramaswamy et al. [27] proved that convex approaches are, in general, inconsistent with the usual loss when ranking from

pairwise preferences, as well as with the Expected Reciprocal Rank [9] and the well-known Average Precision [22], two metrics that are respectively diversity-inducing and diversity-averse [10]. Thus, extending our analysis to general convex risk minimization, two questions remain open: *i) Are there ranking losses for which regression is not consistent, but for which there exists a consistent convex risk minimization approach?* and *ii) When only non-convex surrogate approaches are consistent, is the learning or optimization problem intrinsically more difficult?*

In Section 3, we describe our main result: sorting by expected utilities is optimal if and only if the sublevel sets of the ranking loss are connected, for a suitable notion of connectedness in the space of permutations. This result identifies the fundamental property of ranking losses that is related to the existence of a consistent regression approach. We give an intuitive interpretation of this result in terms of continuity of optimal scoring functions: when regression is not consistent, there necessarily exist data distributions with continuous conditional distributions $x \mapsto P(y|x)$ but such that all optimal scoring functions are discontinuous. On the other hand, when there is a utility function, expected utilities inherit the regularity properties of $x \mapsto P(y|x)$.

In Section 4, we elaborate on the two types of ranking tasks and discuss surrogate approaches. We first answer question *i)* and show that for every ranking loss, whenever a convex risk minimization approach is consistent, then a suitable regression is consistent. Second, we give elements of answer to question *ii)*: when surrogate losses need be non-convex to be consistent, we show that every Lipschitz surrogate loss must have bad local minima.

Our results establish that the general class of convex surrogate losses cannot solve more tasks than plain regression. This clarifies that sophisticated convex approaches for ranking should be justified by better sample complexity more than "better fit" to a specific ranking task, since asymptotically they are either inconsistent or equivalent to a suitable regression. Moreover, the necessary non-global minima of surrogate losses and the discontinuity of optimal scoring are the first formal arguments for the intrinsic difficulties that arise when using score-and-sort for general ranking problems.

## 2 Preliminaries: Learning to rank and Consistency

### 2.1 Learning to Rank

**Supervised learning to rank.** We consider a framework of label or subset ranking [13, 12]. The learner predicts rankings over $n$ items based on input features $x \in \mathcal{X} \subseteq \mathbb{R}^d$, where $\mathcal{X}$ has nonempty interior. Rankings are represented by permutations, and we denote by $\mathfrak{S}_n$ the set of all permutations of $[n] = \{1, \ldots, n\}$. The learner has access to a supervision signal in $\mathcal{Y}$. When the task is fully supervised, $\mathcal{Y} = \mathfrak{S}_n$, but we also allow for weakly supervised settings where a supervision $y \in \mathcal{Y}$ is a vector of relevance judgements for each item, a preference graph, or a partial ranking. Our analysis is agnostic to the type of supervision, we only assume that $\mathcal{Y}$ is finite. The task loss $L : \mathcal{Y} \times \mathfrak{S}_n \to \mathbb{R}$ measures the quality of a ranking given the supervision. The goal is to learn, from supervised training data, a ranking function $h : \mathcal{X} \to \mathfrak{S}_n$ with low *task risk* $\mathcal{R}_{L,P}(h) = \mathbb{E}_P[L(Y, h(X))]$, where the expectation is taken according to the data distribution $P$. We give later examples of usual task losses and their associated $\mathcal{Y}$ in Table 1 (Section 3).

**The score-and-sort approach.** A usual approach to learning to rank is to sort the items by decreasing order of learnt scores. Given a vector of scores $s \in \mathbb{R}^n$, $\mathrm{argsort}(s)$ returns *the set* of permutations that are compatible with a decreasing order of score:

$$\mathrm{argsort}(s) = \{\sigma \in \mathfrak{S}_n : \forall k \in [n-1], s_{\sigma(k)} \geq s_{\sigma(k+1)}\}.$$

We overload $L$ for a set of rankings $\pi \subseteq \mathfrak{S}_n$, using its average value $L(y, \pi) = \frac{1}{|\pi|} \sum_{\sigma \in \pi} L(y, \sigma)$. Thus, given a *scoring function* $f$, i.e., a measurable function $f : \mathcal{X} \to \mathbb{R}^n$, the task risk for the score-and-sort approach is $\mathcal{R}_{L,P}(\mathrm{argsort} \circ f) = \mathbb{E}_P[L(Y, \mathrm{argsort}(f(X)))]$.

Similarly to previous studies on consistency for ranking [e.g., 12, 15, 28, 7, 8, 26], $x$ contains the information about the input and all items. This is the natural setup when ranking class labels in a multiclass/multilabel setting, since in that case there are only input features (e.g., an image). In recommender systems or search engines, this means that we allos the score of an item to depend on the other available items. This setup makes sure that every ranking function can be implemented by the score-and-sort approach. The difficulty of learning to rank comes from the requirement to learn the scoring function from noisy supervision, instead of from a deterministic ground-truth ranking.

## 2.2 Consistency, Calibration and Utilities

**Calibration of surrogate losses and consistency.** In the score-and-sort approach, the scoring function is not trained by minimizing $\mathcal{R}_L(\mathrm{argsort} \circ f)$ (or its empirical counterpart) because it is computationally hard in general. Rather, a *surrogate loss* is used, which is a measurable function $\Phi : \mathcal{Y} \times \mathbb{R}^n \to \mathbb{R}_+$. Learning algorithms aim at minimizing the *surrogate risk*, defined for a scoring function $f : \mathcal{X} \to \mathbb{R}^n$ as $\mathcal{R}_{\Phi,P}(f) = \mathbb{E}_P[\Phi(Y, f(X))]$. We analyze the consistency of surrogate risk minimization, which informally states that minimizing $\mathcal{R}_{\Phi,P}(f)$ over $f$ leads to minimizing $\mathcal{R}_{L,P}(\mathrm{argsort} \circ f)$. More formally, an *excess risk bound* between $\mathcal{R}_\Phi$ and $\mathcal{R}_L$ is a continuous function $\delta : \mathbb{R}_+ \to \mathbb{R}_+$ with $\delta(\epsilon) \xrightarrow[\epsilon \to 0]{} 0$ such that, for every distribution $P$ over $\mathcal{X} \times \mathcal{Y}$:

$$\forall f : \mathcal{X} \to \mathbb{R}^n, \quad \mathcal{R}_{L,P}(\mathrm{argsort} \circ f) - \inf_{h:\mathcal{X} \to \mathfrak{S}_n} \mathcal{R}_{L,P}(h) \le \delta\Big(\mathcal{R}_{\Phi,P}(f) - \inf_{g:\mathcal{X} \to \mathbb{R}^n} \mathcal{R}_{\Phi,P}(g)\Big).$$

where we implicitly restrict to measurable functions. Excess risks bounds give distribution-independent guarantees on the convergence of the task risk given the convergence of the surrogate risk to its infimum. The guarantees are asymptotic in nature, because the infimum over all measurable functions of the risks are, in general, approachable only as the number of samples tends to infinity.

Following Steinwart [31], we study consistency through *inner risks* and *calibration*. Let $\mathcal{Q}$ be the set of probability mass functions over $\mathcal{Y}$. Given $q \in \mathcal{Q}$ and $\psi : \mathcal{Y} \to \mathbb{R}$, let $\mathbb{E}_q[\psi(Y)] = \sum_{y \in \mathcal{Y}} q(y)\psi(y)$. The inner task risk $\ell : \mathcal{Q} \times \mathfrak{S}_n \to \mathbb{R}$ and the inner surrogate risk $\phi : \mathcal{Q} \times \mathbb{R}^n \to \mathbb{R}_+$ are:

$$\ell(q, \sigma) = \mathbb{E}_q[L(Y, \sigma)] \qquad \phi(q, s) = \mathbb{E}_q[\Phi(Y, s)] \qquad \underline{\phi}(q) = \inf_{s \in \mathbb{R}^n} \phi(q, s).$$

The terminology "inner risk" is justified by the equality $\mathcal{R}_{L,P}(h) = \mathbb{E}_P[\ell(P(.|X), h(X))]$, which holds for every $h : \mathcal{X} \to \mathfrak{S}_n$ and where $x \mapsto P(.|x)$ is the conditional distribution of $Y$ given $X = x$. Similarly to $L$, we extend $\ell$ to subsets $\pi \subseteq \mathfrak{S}_n$ using an average: $\ell(q, \pi) = \frac{1}{|\pi|} \sum_{\sigma \in \pi} \ell(q, \sigma)$..

Our definition of calibration is adapted from [31, Def 2.7] to the case where $\mathcal{Y}$ and $\mathfrak{S}_n$ are finite:

**Definition 1.** *Given $L : \mathcal{Y} \times \mathfrak{S}_n \to \mathbb{R}$, the surrogate loss $\Phi : \mathcal{Y} \times \mathbb{R}^n \to \mathbb{R}_+$ is $L$-calibrated if*

$$\forall q \in \mathcal{Q}, \exists \delta > 0, \forall s \in \mathbb{R}^n, \quad \phi(q, s) - \underline{\phi}(q) \le \delta \implies \mathrm{argsort}(s) \subseteq \operatorname*{argmin}_{\sigma \in \mathfrak{S}_n} \ell(q, \sigma).$$

Since $\mathcal{Y}$ is finite, using the notation of Def 1, we have (see [31, Th. 2.8] and Appendix B):

**Proposition 2.** *$\Phi$ is $L$-calibrated if and only if there is an excess risk bound between $\mathcal{R}_\Phi$ and $\mathcal{R}_L$.*

**Consistency of regression and utility functions.** We first study a problem equivalent to the consistency of square loss regression. In general, the supervision $y$ does not necessarily contain suitable target scores for the regression, so we use intermediate utility values. Given a task loss $L : \mathcal{Y} \times \mathfrak{S}_n \to \mathbb{R}$, we say that $u : \mathcal{Y} \to \mathbb{R}^n$ is a *a utility function* for $L$ if the square loss $\Phi_u^{\mathrm{sq}}(y, s) = (s - u(y))^2$ is $L$-calibrated. We say that $L$ is *compatible with expected utility* (CEU) if a utility function exists for $L$. Denoting $\phi_u^{\mathrm{sq}}$ the inner risk of $\Phi_u^{\mathrm{sq}}$, we have $\operatorname{argmin}_s \phi_u^{\mathrm{sq}}(q, s) = \{\mathbb{E}_q[u(Y)]\}$, so that

$$L \text{ is CEU} \iff \exists u : \mathcal{Y} \to \mathbb{R}^n, \forall q \in \mathcal{Q}, \mathrm{argsort}\Big(\mathbb{E}_q[u(y)]\Big) \subseteq \operatorname*{argmin}_{\sigma \in \mathfrak{S}_n} \ell(q, \sigma).$$

The characterization of CEU task losses for ranking is our main technical result, presented in Section 3. We extend the analysis to general surrogate losses in Section 4.

## 2.3 Ranking losses

An arbitrary function $\mathcal{Y} \times \mathfrak{S}_n \to \mathbb{R}$ is not necessarily a valid task loss for ranking. The definition of a *ranking loss* below isolates minimal properties that make a function $\mathcal{Y} \times \mathfrak{S}_n \to \mathbb{R}$ suitable to specify a ranking task. We use $\tau_{ij}$ to denote the transposition that swaps items $i$ and $j$, i.e., for $i' \in [n]$, $\tau_{ij}(i') = i'$ if $i' \notin \{i, j\}$, $\tau_{ij}(i) = j$ and $\tau_{ij}(j) = i$. Notice that given the definition of $\mathrm{argsort}$ in (1), lower ranks are better, $\sigma(k)$ is the item at rank $k$ and $\sigma^{-1}(i)$ is the rank of item $i$.

**Definition 3** (Ranking loss). *A task loss $L : \mathcal{Y} \times \mathfrak{S}_n \to \mathbb{R}$ is a* ranking loss *if its inner risk satisfies*

- *Items are equivalent* a priori*: $\forall q \in \mathcal{Q}, \forall i, j \in [n], \exists q' \in \mathcal{Q}$ s.t. $\forall \sigma \in \mathfrak{S}_n, \ell(q, \sigma) = \ell(q', \tau_{ij}\sigma)$.*

- *One distribution over $\mathcal{Y}$ strictly prefers item $i$ over other items:* $\quad \forall i \in [n], \exists q_{\text{top}}^{(i)} \in \mathcal{Q}$ *s.t.:*

  1. $\ell(q_{\text{top}}^{(i)}, .)$ *strictly decreases with the rank of $i$:*

$$\forall \sigma, \nu \in \mathfrak{S}_n, \left( \sigma^{-1}(i) < \nu^{-1}(i) \Rightarrow \ell(q_{\text{top}}^{(i)}, \sigma) < \ell(q_{\text{top}}^{(i)}, \nu) \right) ,$$

  2. *Ranks of items other than $i$ do not matter:*

$$\forall \sigma \in \mathfrak{S}_n, \forall j \neq i \in [n], \forall j' \neq i \in [n], \ \ell(q_{\text{top}}^{(i)}, \sigma) = \ell(q_{\text{top}}^{(i)}, \tau_{jj'}\sigma) .$$

The first point of the definition is always satisfied in practice since indexes of items do not carry any meaning. In usual ranking tasks, $q_{\text{top}}^{(i)}$ is trivial. For instance, when the supervision is a vector of relevance judgements, $q_{\text{top}}^{(i)}$ is a Dirac on $y$ that gives a high relevance to $i$ and the same lower relevance to all other items. The only practical restriction of this definition is the requirement of *strict* improvement in the definition of $q_{\text{top}}^{(i)}$. It is satisfied by some notions of DCG, the AP and the ERR, but not by task losses for $K$-subset selection and top-$K$ ranking tasks (in these tasks, ranks of $i$ larger than $K$ lead to the same loss). In the appendix, we extend all our results to a more general case that captures selection and top-$k$ ranking tasks (see Appendix A for the general setup). In the main paper, we focus on ranking only to keep the exposition simple.

For ranking losses, calibration is an "equality of argmins" instead of an inclusion:

**Theorem 4** ([7, Th. 2]). *Given a ranking loss $L$, $\Phi : \mathcal{Y} \times \mathbb{R}^n \to \mathbb{R}_+$ is $L$-calibrated if and only if*

$$\forall q \in \mathcal{Q}, \exists \delta_0 > 0, \forall \delta \in (0, \delta_0], \quad \underset{\sigma \in \mathfrak{S}_n}{\operatorname{argmin}} \ell(q, \sigma) = \bigcup_{s: \phi(q,s) - \underline{\phi}(q) < \delta} \operatorname{argsort}(s)$$

This equality means that for $L$-calibrated surrogate losses, properties of sublevel sets of the inner risk $isurrogate$ translate into similar properties on $\ell$ and vice-versa. The next section characterizes the sublevel sets of ranking losses that are CEU, while Section 4 focuses on calibrated surrogate losses.

## 3 Utility functions and connectedness

This section studies sublevel sets of inner ranking risks in terms of their connectedness:

**Definition 5.** *A set $\pi \subseteq \mathfrak{S}_n$ is* connected *if there is a connected set $S \subseteq \mathbb{R}^n$ s.t. $\pi = \bigcup_{s \in S} \operatorname{argsort}(s)$.*

The relationship with topological notion of connectedness is given in Appendix A.

In our main result below, we denote by $\operatorname{lev}_\epsilon \ell(q, .)$ the strict $\epsilon$-sublevel set of the excess inner risk at $q$: $\operatorname{lev}_\epsilon \ell(q, .) = \{\sigma : \ell(q, \sigma) - \min_{\sigma'} \ell(q, \sigma') < \epsilon\}$.

**Theorem 6.** *For a ranking loss $L$, the following statements are equivalent:*

  *(i) $L$ is CEU,*

  *(ii) $\forall q, \operatorname{argmin}_\sigma \ell(q, \sigma)$ is connected.*

  *(iii) $\forall \epsilon > 0, \forall q, \operatorname{lev}_\epsilon \ell(q, .)$ is connected,*

*Moreover, the function $\tilde{u} : \mathcal{Y} \to \mathbb{R}^n$ defined as:* $\quad \forall i \in [n], \tilde{u}_i(y) = - \sum_{\sigma \in \mathfrak{S}_n} \mathbb{1}_{\{\sigma(1)=i\}} L(y, \sigma)$

*is a utility function for $L$ whenever there exists a utility function for $L$ (i.e., whenever $L$ is CEU).*

The proof is given in Appendix C. The theorem has four parts: the existence of the utility function, the connected argmins, and the connected sublevel sets, which in turn give an explicit formula for a utility function for $L$ when one exists. We discuss these four aspects in more detail below.

### 3.1 Disconnected argmins and discontinuity of optimal scoring functions

While the connectedness of argmins and of sublevel sets is the fundamental concept underlying our result, we first give a more intuitive interpretation of the connectedness of argmins (point *ii)* of Th. 6) in terms of the continuity of optimal predictors. The full proof is in Appendix D.

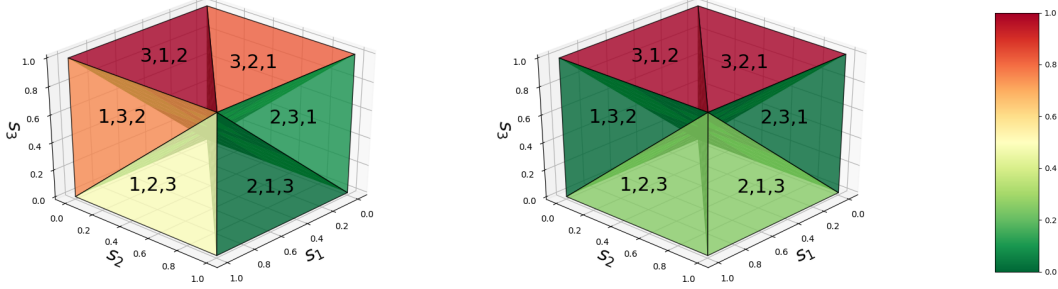

Figure 1: Graphical representation of level sets of the DCG (left) and the ERR (right) for three items. (normalized in [0,1], lower is better). For the DCG, the distribution over $\mathcal{Y}$ is a Dirac on $(1, 2, 0)$. For the ERR it is a mixture of a Dirac on $(1, 1, 0)$ and on $(0, 0, 1)$ with weights $\frac{7}{12}$ and $\frac{5}{12}$. Each axis is the score of one item, and each region is the corresponding argsort. Two regions share an edge (i.e., are adjacent)if the permutations differ only by a transposition of adjacent items. A subset of regions is *connected* if it is possible to stay in that subset while moving from one region to the other using a path of adjacent regions. The sublevel sets for the DCG are all connected, but for the ERR, the sublevel set highlighted in dark green has two connected components.

**Corollary 7.** *A ranking loss $L$ is CEU if and only if: for every distribution $P$ over $\mathcal{X} \times \mathcal{Y}$ such that $x \mapsto P(.|x)$ is continuous, there is a* continuous *optimal scoring function for $\mathcal{R}_{L,P}$.*

The result follows from the preservation of connectedness by continuous functions. If a ranking loss has a utility function $u$, an optimal scoring function is $x \mapsto \mathbb{E}_{P(.|x)}[u(Y)]$, which inherits all the regularity properties of $x \mapsto P(.|x)$ since $\mathcal{Y}$ is finite. The value of Cor. 7 is thus to show that *only* CEU losses always have continuous optimal scoring functions when $x \mapsto P(.|x)$ is continuous.

*Sketch of proof.* The "only if" direction is straightforward. The full proof of the "if" direction is deferred to the appendix. The main line is the following: take a ranking loss $L$ such that there is a continuous optimal scoring function for every distribution such that $P(.|x)$ is continuous. Aiming for a contradiction, assume that $L$ is not CEU. Then, by Theorem 6, there is a $q$ such that $\operatorname{argmin}_\sigma \ell(q, \sigma)$ is disconnected. Taking $\mathcal{X} = [0, 1]$ without loss of generality and using a uniform marginal distribution over $\mathcal{X}$, we construct a conditional distribution $P(.|x)$ that continuously goes between two distributions $q_0$ and $q_1$, where each one of $\operatorname{argmin}_\sigma \ell(q_0, \sigma)$ and $\operatorname{argmin}_\sigma \ell(q_1, \sigma)$ are included in different connected components of $q$. In that construction, $q$ itself is used as an intermediate point between $q_0$ and $q_1$ to make sure that $x \mapsto P(.|x)$ is continuous. If this distribution over $\mathcal{X} \times \mathcal{Y}$ had a continuous optimal scoring function, then this continuous function would connect two distinct connected components of $\operatorname{argmin}_\sigma \ell(q, \sigma)$, which is a contradiction. □

This interpretation of Theorem 6 in terms of continuous optimal predictors indicates that CEU and non-CEU losses lead to learning problems of different intrinsic difficulty. We elaborate more on this dichotomy between CEU and non-CEU ranking losses in the next subsection and in Section 4.

### 3.2 Disconnected sublevel sets and bad local minima

We now interpret the connectedness of sublevel sets in terms of local minima of the ranking loss. A permutation is a local minimum if exchanging two adjacent items only increases the loss:

**Definition 8.** *Given a distribution $q \in \mathcal{Q}$ and a loss $L$, a ranking $\sigma \in \mathfrak{S}_n$ is a local minimum if for any $r \in [n-1]$, $\ell(q, \sigma) \leq \ell(q, \sigma\tau_{r,r+1})$.*

The relationship with the connectedness of sublevel sets is more apparent with this characterization of connectedness in $\mathfrak{S}_n$ (the proof is in Appendix C):

**Proposition 9.** *A set $\pi \subseteq \mathfrak{S}_n$ is connected if and only if for every $\sigma, \nu \in \pi$ there is path between $\sigma$ and $\nu$ in $\pi$ using transpositions of adjacent items, i.e. $\exists \sigma_0 = \sigma, \sigma_1, ..., \sigma_M = \nu$ where $\forall m, \exists k : \sigma_{m+1} = \sigma_m \tau_{kk+1}$ and $\forall m, \sigma_m \in \pi$.*

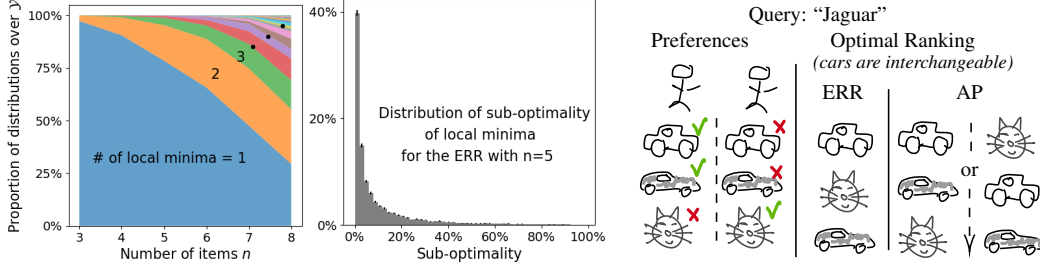

Figure 2: *(left)* Proportions of distributions $q \in \mathcal{Q}$ *(y*-axis) vs number of local minima (colors) vs number of items (*x*-axis) for the ERR. For $n = 8$, nearly 70% of distributions have several local minima. *(middle)* distribution of sub-optimality of the sub-optimal local minima for the ERR for $n = 4$. The sub-optimality of a local minimum is $\frac{\text{value}-\min}{\max-\min}$ in %. 0.25% of local minima are more than 90% sub-optimal. *(right)* Illustration of optimal rankings for the ERR (diversity-inducing) and the AP (diversity-averse), for the fictional search engine scenario with the ambiguous query "jaguar".

That is, in a connected set $\pi$, we can go from one ranking to another by iteratively swapping adjacent items (as in the bubble sort algorithm), while staying in $\pi$. A gaphical representation of connected and disconnected sets is given in Figure 1.

When $L$ is CEU, then by point *iii)* of Th. 6, all sublevel sets are conected, which means that for any ranking in a sublevel set, we can find a path using transpositions of adjacent items to an optimal ranking that never increases the loss. Thus, for CEU losses, all local minima are global minima.

Conversely, when $L$ is not CEU, then for some distributions $q$, the inner ranking risk $\ell$ has disconnected sublevel sets, leading to bad local minima. We show in Appendix E that the set $\{q \in \mathcal{Q} : q$ has at least one non-global local minimum$\}$ has non-zero Lebesgue measure for all such $L$. In order to give concrete numbers, we consider the Expected Reciprocal Rank (ERR) [9], which is not CEU as we see later in Section 3.3. By uniformly sampling $q \in \mathcal{Q}$, we empirically estimated proportions of distributions $q$ vs number of local minima for $\ell(q,.)$, for different numbers of items $n$. The results are plotted in Fig. 2 *(left)*. The probability that $\ell(q,.)$ has several local minima increases rapidly with $n$, which is expected since when $n$ is small a larger proportion of permutations are adjacent to each other. For $n = 8$, we found that the inner risk has up to 23 local minima, with several local minima for nearly 70% of distributions. Fig. 2 *(middle)* displays the sub-optimality of these local minima, showing that 25% of them are more than 10% sub-optimal. Appendix E contains more details and presents additional results for the Average Precision that are qualitatively similar.

The absence of local minima in $L$ when $L$ is CEU echoes the absence of local minima of the square loss $\Phi_u^{\text{sq}}$ that is $L$-calibrated. The result is far from trivial though, because calibration is only a property of minimizers of the inner ranking risk. The strength of our result is to understand that connectedness of argmins actually implies connectedness of all sublevel sets.

### 3.3 Utilities, diverse rankings and disconnected argmins

We now discuss Theorem 6 in light of prior works. First, it is well-known [12, 6, 28] that gain-based metrics of the form $\text{DCG}_{w,u}(y, \sigma) = -\sum_{k=1}^{n} w_k u_{\sigma(k)}(y)$, which include the Discounted Cumulative gain or precision/recall at $K$ have $u$ as a utility function. We provide example formulas for well-known ranking losses in Table 1, more examples can be found in [see e.g., 8, Table 1]. We show in Appendix F that $u$ is equal to $\tilde{u}$ given in Th. 6 up to an affine transformation.

Conversely, it is also known that the ERR and the AP are not CEU, because of their non-neutral behavior with respect to diverse rankings [6, 7, 26]. To illustrate the effect of diversity on connectedness of the argmins, we use the examples of Calauzènes et al. [7, Table 2]. Let us consider a fictional search engine scenario depicted in Figure 2 (right), in which the ambiguous query "jaguar" is interpreted differently by two annotators (e.g., as the animal, or as the car brand) leading to noise in the supervision. Optimal rankings for a diversity-inducing loss (e.g., the ERR) alternate items relevant for each interpretation, while for a diversity-averse loss (e.g., the AP), they cluster items relevant to the same interpretation. In both cases, we cannot find a path between optimal rankings by swapping adjacent items without leaving the argmin, so the argmin is not connected.

| $\mathcal{Y}$ | Loss | Formula | $u_i(y)$ |
|---|---|---|---|
| $y \in \{0,1\}^n$ | Prec@K | $-\sum_{k=1}^{K} \frac{y_{\sigma(k)}}{K}$ | $y_i$ |
| | AP | $\frac{-1}{\|y\|_1} \sum_{i:y_i=1} \text{Prec@}\sigma^{-1}(\text{i})(\text{y},\sigma)$ | $\times$ |
| | AUC | $\sum_{\substack{i:y_i=1\\j:y_j=0}} \frac{\mathbb{1}_{\{\sigma^{-1}(i)>\sigma^{-1}(j)\}}}{\|y\|_1(n-\|y\|_1)}$ | $\frac{y_i}{\|y\|_1(n-\|y\|_1)}$ |
| $y \in \{0,\ldots,p\}^n$ | DCG@K | $-\sum_{k=1}^{K} \frac{2^{y_{\sigma(k)}}-1}{\log(1+k)}$ | $2^{y_i}-1$ |
| | NDCG@K | $\frac{-\text{DCG@K}(y,\sigma)}{\max\limits_{\nu\in\mathfrak{S}_n} \text{DCG@K}(y,\nu)}$ | $\frac{2^{y_i}-1}{\max\limits_{\nu\in\mathfrak{S}_n} \text{DCG@K}(y,\nu)}$ |
| | ERR | $-\sum_{k=1}^{n} \frac{R_{\sigma(k)}}{k} \prod_{r=1}^{k-1}(1-R_{\sigma(r)}), \quad R_i=\frac{2^i-1}{2^p}$ | $\times$ |
| $y \in \text{DAG}_n$ | PD | $\sum_{i\to j\in y}\mathbb{1}_{\{\sigma^{-1}(i)>\sigma^{-1}(j)\}}$ | $\times$ |
| $y \in \mathfrak{S}_n$ | Spearman | $\frac{6}{n(n^2-1)}\sum_{i=1}^{n}(\sigma^{-1}(i)-y^{-1}(i))^2-1$ | $n-y^{-1}(i)$ |

Table 1: Example of ranking losses with their utilities, if any. We give examples with different types of supervision, including $\text{DAG}_n$, which is the set of directed acyclic graphs used in the computation of the pairwise disagreement (PD) studied by Duchi et al. [15].

The works above studied convex approaches for ranking. As we explain next, compared to these works, the value of Th. 6 is to show that only CEU ranking losses have a convex, calibrated surrogate.

## 4 Calibrated Surrogate Losses

When a ranking loss $L$ has a utility function $u$, the square loss $\Phi_u^{\text{sq}} : y, s \mapsto (s - u(y))^2$ is $L$-calibrated. Many other convex surrogate losses are applicable when a utility function is available: losses based on a general Bregman divergence $D$ of the form $y, s \mapsto D(\tilde{u}(y), \psi(s))$ where $\psi$ is a link function [28], pointwise losses such as $y, s \mapsto \sum_i \tilde{u}_i(y) \log(1 + e^{-s_i}) + \lambda \sum_i \log(1 + e_i^s)$ where $\lambda > 0$ is a hyperparameter, as well as pairwise losses such as $y, s \mapsto \sum_{i,j} \tilde{u}_i(y) \log(1 + e^{s_j - s_i})$ [8]. These losses are *convex*, in the sense that $\Phi : \mathcal{Y} \times \mathbb{R}^n \mathbb{R}_+$ is convex when every $y \in \mathcal{Y}$, the function $s \mapsto \Phi(y, s)$ is convex. Also, for all of them, the minimizers of the inner risk are equal to expected utilities up to a strictly monotonic transform. These losses are well understood, and come with numerous guarantees, such as explicit excess risk bounds for gain-based losses [12, 28, 8]. There are also fast rates of estimation of the scoring function when $x \mapsto P(.|x)$ is smooth [see e.g., 4].

Our main result above, Theorem 6 implies that whenever there exists a convex, calibrated loss, then these utility-based surrogate losses are calibrated (if given the suitable utility function):

**Theorem 10.** *Given a ranking loss $L$, there is an $L$-calibrated* convex *loss if and only if $L$ is CEU.*

*Proof of Theorem 10.* The reverse implication is straightforward: if $L$ admits a utility $u$, then $y, s \mapsto (s - u(y))^2$ is convex and $L$-calibrated. For the direct implication, let $q \in \mathcal{Q}$. Since $\phi(q, .)$ is convex, its sublevel sets are all convex. Thus, if $\Phi$ is $L$-calibrated, then $\operatorname{argmin}_\sigma \ell(q, \sigma)$ is generated by a convex (and thus connected) set of scores by Theorem 4. By the definition of connectedness (Def. 5) $\operatorname{argmin}_\sigma \ell(q, \sigma)$ is connected. This holds for all $q$, so $L$ admits a utility function by Theorem 6. $\square$

This result establishes the fundamental role of utilities and regression in learning to rank with convex risk minimization: for non-CEU ranking losses, only non-convex surrogate losses are calibrated. For instance, it implies that sophisticated convex approaches such as structural SVMs for ranking [see e.g., 21, 39] do not asymptotically solve more tasks than regression.

## 4.1 The case of convex surrogate losses

Th. 10 closes the question of the calibration of convex losses for ranking. The question was initiated by Duchi et al. [15, Section 2.1] in the context of learning from pairwise preferences, and then extended to general ranking and classification losses [7, 26].

Calauzènes et al. [7] proved that the AP, the ERR and the pairwise disagreement cannot have calibrated, convex surrogate losses because $\operatorname{argmin}_\sigma \ell(q, ., \sigma)$ is not connected for specific choices of $q$. Their result is essentially *i)* $\implies$ *ii)* of our Theorem 6. By proving the reverse implication, we show that *the general class of convex surrogate losses is not more general than square loss regression.* Notice that *i)* $\implies$ *ii)* is an immediate consequence of Th. 4 (also [7, Theorem 2]), while proving the reverse implication is technically much more challenging.

Ramaswamy and Agarwal [26] defined the concept of *convex calibration dimension*. On the one hand, their approach is more general and they study the minimum number of degrees of freedom that are required for a convex approach to be calibrated. However, they use an unstructured inference procedure, which in general does not correspond to sorting and may be computationally hard. Thus, they do not study when convex losses are calibrated with a score-and-sort approach.

## 4.2 When no convex surrogate loss is calibrated

For non-CEU ranking losses, Th. 10 shows that $L$-calibrated surrogate losses are necessarily non-convex. Additionally, we explained in Sections 3.1 and 3.2 that non-CEU ranking losses have a complex landscape with local minima and discontinuous optimal scoring functions. We now describe analogous undesirable properties for their calibrated (non-convex) surrogate losses.

To illustrate the claims of this section, we perform simulations using a non-convex surrogate loss defined by smoothing the task loss, similarly to [33, 35, 20]. We follow the idea of the probit loss for classification [18], which is a Gaussian smoothing of the 0/1 loss. Here, we use a Gumbel kernel $\kappa$ instead of a Gaussian kernel, because the resulting smoothed loss has a closed form formulation using the Plackett-Luce model [38], and is always $L$-calibrated (a proof is given in Appendix G).

**Proposition 11.** *For any ranking loss $L$, the following surrogate loss $\Phi_L^{\mathrm{NC}}$ is $L$-calibrated.*

$$\Phi_L^{\mathrm{NC}} : y, s \mapsto \int_{\mathbb{R}^n} L(y, \operatorname{argsort}(u-s)) \kappa(u) \mathrm{d}u = \sum_{\sigma \in \mathfrak{S}_n} L(y, \sigma) \prod_{r=1}^n \frac{e^{s_{\sigma(r)}}}{\sum_{k \geq r} e^{s_{\sigma(k)}}}$$

**Existence of sub-optimal local minima.** Th. 12 below is the counterpart of Section 3.2 for surrogate losses. In essence, it shows that when $L$ is not CEU, then bounded, Lipschitz surrogate losses that are $L$-calibrated have bad local minima. In order to deal with surrogate losses like $\Phi_L^{\mathrm{NC}}$ where critical points are at infinity, we use the following generalization of local minimum. Let $\operatorname{lev}_\epsilon \phi(q, .)$ denote the strict $\epsilon$-sublevel set of the excess inner (surrogate) risk: $\operatorname{lev}_\epsilon \phi(q, .) = \{s \in \mathbb{R}^n : \phi(q, s) - \underline{\phi}(q) < \epsilon\}$. Given $\epsilon > 0$, a *local valley* is a connected component of $\operatorname{lev}_\epsilon \phi(q, .)$. A *bad local valley* is a local valley $C$ such that $\inf_{s \in C} \phi(q, s) > \underline{\phi}(q)$ [23, Def. 4.1].

**Theorem 12.** *Let $L$ be a non-CEU ranking loss. Let $\Phi : \mathcal{Y} \times \mathbb{R}^n \to \mathbb{R}_+$ such that $s \mapsto \Phi(y, s)$ is bounded and Lipschitz for every $y \in \mathcal{Y}$. If $\Phi$ is $L$-calibrated, then the set $\{q \in \mathcal{Q} : \phi(q, .) \text{ has bad local valleys}\}$ has non-zero Lebesgue measure.*

To give more precise figures on these local valleys, Fig. 3 *(left and middle)* shows the distribution of sub-optimalities of bad local valleys for the surrogate $\Phi_L^{\mathrm{NC}}$ applied to the ERR and the AP. These local valleys are found by gradient descent, starting from a relatively large random initialization to analyze the global loss surface. The distributions $q$ are uniformly sampled over $\mathcal{Q}$, rejecting the distributions $q$ where $\ell(q, .)$ does not have any local minima, as for Fig. 2 *(middle)*. Fig. 3 *(right)* show the proportions or runs on these distributions that end up stuck in a bad local valley when using an initialization close to 0 (which empirically was best to avoid bad local valleys). As for the ranking losses, we see that the surrogate losses also have "bad local minima", but also that gradient descent algorithms might be stuck in them. Details of these experiments are given in Appendix H.

**(Bad) approximations of optimal scoring functions by Lipschitz functions.** In Section 3.1, we showed that optimal scoring functions for non-CEU ranking losses are discontinuous for some

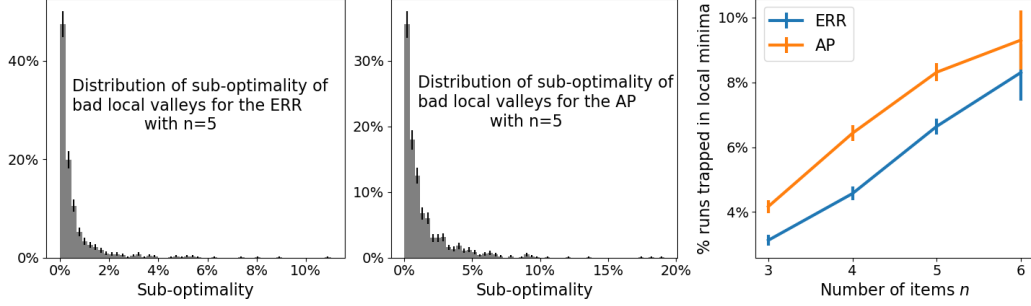

Figure 3: *Left and middle* Distributions of sub-optimalities of local valleys for the surrogate $\Phi_L^{\text{NC}}$ applied to the ERR and the AP on random distributions where the ERR/the AP have bad local minima. *(right)* percentage of optimization runs based on gradient descent that end up stuck in local minima.

distributions where $x \mapsto P(.|x)$ is continuous. Prop. 13 below is a stronger analogous statement for surrogate losses when $x \mapsto P(.|x)$ is Lipschitz. The proof can be found in Appendix H.

**Proposition 13.** *Let $L$ be a non-CEU ranking loss. Let $\Phi : \mathcal{Y} \times \mathbb{R}^n \to [0, B_\Phi]$ be an $L$-calibrated loss such that $\Phi(y, .)$ is $\beta_\Phi$-Lipschitz for all $y \in \mathcal{Y}$.*

*Then, there exists constants $c, c' > 0$ and a probability measure $P$ over $[0, 1] \times \mathcal{Y}$ where $x \mapsto P(.|x)$ is Lipschitz, such that for all $\beta \geq 0$:*

$$\inf_{\substack{f:[0,1]\to\mathbb{R}^n \\ f:\beta-Lipschitz}} \mathcal{R}_{\Phi,P}(f) - \inf_{g:\mathcal{X}\to\mathbb{R}^n} \mathcal{R}_{\Phi,P}(g) \geq \min\left(c', \frac{c}{8B_\Phi + \beta_\Phi\beta}\right).$$

Notice that for CEU ranking losses with utility function $u$, denoting $\beta_P$ the Lipschitz coefficient of $x \mapsto P(.|x)$, the optimal scoring function is $\|u\|_\infty \beta_P$-Lipschitz. In contrast, for non-CEU losses, the lower bound above shows that for some distributions, the Lipschitz constant of the scoring function $\beta$ needs to grow to infinity to minimize a Lipschitz, calibrated surrogate loss.

## 5 Dicussion and conclusion

For supervised ranking with the score-and-sort approach, learning the scoring function through regression is consistent for all ranking tasks for which a convex risk minimization approach is consistent. When regression is not consistent, surrogate risks have bad local minima and their minimizers cannot be approximated by regular functions even when $P(.|x)$ is regular. These results demonstrate the fundamental role of regression among convex methods for ranking. They also cast light on the undesirable properties of the score-and-sort approach for non-CEU ranking losses.

For tasks with non-CEU ranking losses, one possible avenue is to develop efficient direct loss minimization approaches, such as approximations of $\Phi_L^{\text{NC}}$ above or as proposed by Song et al. [30]. Another direction is to find alternatives to score-and-sort. Ramaswamy and Agarwal [26] developed a more general approach which allows for consistent convex approaches, but in higher dimension than the number of items. Excess risk bounds follow from the work on stuctured prediction [11, 25]. The drawback of these approaches is that inference might be NP-hard [15, 26]. In multilabel classification, in some rare cases, efficient inference procedures have been found [36], but not yet in ranking. A third direction is to relax the requirement of asymptotic optimality. By choosing a sensible but efficient inference procedure and making additional assumptions on the data distribution. Chapelle et al. [10] followed this approach for divesifying search results. For now, the theoretical properties of for such approaches have not been investigated. A possible starting point would be to build on the recent work on excess risk bounds for non-calibrated losses [32].

## Broader Impact

The framework of ranking we studied plays a fundamental role in information retrieval and information filtering systems. While these systems play an undeniable positive role in society because they increase the efficiency of information access, they also shape the landscape of what their users get to know. This lead to questions regarding equal representation in search engines result [34], how they represent social groups [17], and whether they exacerbate filter bubbles or act as echo chambers [3]. As the most recent trends in learning to rank involve randomized experiments and online learning [16, 1] to improve quality, such user experiments also need be carried out with care [5].

Creating a strong theory of learning to rank is important to address the challenges of information access systems. For instance, methods to produce diverse rankings might be part of the solution, and for now the theory of machine learning for diverse rankings is scarce. This paper partly fills this gap.

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
