[Supplementary Material]

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

## Footnotes

[1] proof: $\tau_{ij}\Lambda(\sigma) \in \mathcal{Z}$ because $\mathcal{Z}$ is closed by transpositions. By the definition of $\tau_{ij}z$ for $z \in \mathcal{Z}$, $\tau_{ij}\Lambda(\sigma)$ must contain $\tau_{ij}\sigma$. Since $\mathcal{Z}$ is a partition of $\mathfrak{S}_n$, there is a unique element of $\mathcal{Z}$ that contains $\tau_{ij}\sigma$, which is, by definition of $\Lambda$, $\Lambda(\tau_{ij}\sigma)$.

[2]In that definition, we consider an item to be adjacent to itself. We could prevent that without altering anything in the proofs.

[3]Note that even though tie breaking lemmas are purely technical, they are very important because connectedness is materialized by ties. There are two ways to break ties: the tie breaking lemma chooses the argmin, but cannot guarantee that other ties outside the argmin are suitably broken. On the other hand, Lemma 39 breaks a specific tie, but cannot choose precisely the resulting argmin.

[4]The last equality comes from noticing $\underline{\ell}(\bar{q}(\alpha_0)) = \ell(\bar{q}(\alpha_0), z)$ and decomposing

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

# A General Setting

*Disclaimer: As mentioned in the body of the paper, we address, in this Appendix, a setting more general than ranking. We pay attention to often reinterpret the results with the ranking setting described in the paper to ease the reader's task.*

In supervised learning, a learner has access to input features $x \in \mathcal{X} \subseteq \mathbb{R}^d$ and wants to predict output variables $z \in \mathcal{Z}$ (finite). To do so, she can learn a prediction function $h : \mathcal{X} \to \mathcal{Z}$, belonging to the space of measurable functions from $\mathcal{X}$ to $\mathcal{Z}$ denoted $\mathcal{H}$, using some feedback $y \in \mathcal{Y}$ (finite) and minimizing a task loss $L : \mathcal{Y} \times \mathcal{Z} \to \mathbb{R}$ in expectation over some joint distribution $P$ on the observable variables:

$$\inf_{h \in \mathcal{H}} \mathcal{R}_{L,P}(h) \qquad \text{where } \mathcal{R}_{L,P}(h) = \mathbb{E}_P \left[ L(Y, h(X)) \right]$$

Note that the infimum exists because $L$ is bounded (because $\mathcal{Z}$ and $\mathcal{Y}$ being finite). We denote by $\|L\|_\infty = \max_{y,z} |L(y,z)|$.

**Prediction space: $\mathcal{Z}$ vs $\mathfrak{S}_n$.** Intuitively, when predictions are ranking $\sigma \in \mathfrak{S}_n$, they cannot encode *indifference* between items (the ordering is strict). However, all the results still holds when items can be indifferent (the ordering is a weak order). Further, moving from total orders to weak orders allows to extends the range of applications. For instance, multiclass classification can be seen as predicting a weak order amongst those that strictly prefers one item (label) to all the other items, that are indifferent one from each other. Top-k ranking and subset selection can be handled similarly. The prediction space $\mathcal{Z}$ allows to handle the additional formalism (detailed below) required to handle this generalization.

**Two notes on notation.** For $n \in \mathbb{N}$, we denote $[n]$ the set of integers going from 1 to $n$. The indicator function is denoted $\mathbb{1}_{[.]}$. For $s \in \mathbb{R}^n$ and $\varepsilon > 0$, $\mathcal{B}_2(s, \varepsilon) = \{s' \in \mathbb{R}^n, \|s' - s\|_2 < \varepsilon\}$. Similarly, $\mathcal{B}_\infty(s, \varepsilon)$ denotes the open ball of infinity norm. For $q \in \mathcal{Q}$, the $\|.\|_1$ ball in $\mathcal{Q}$ is denoted by $\mathcal{B}_1(q, \varepsilon) = \{q' \in \mathcal{Q} : \|q - q'\|_1 < \varepsilon\}$. Random variable are uppercase version of their realisation counterparts. Given a function $g : \mathcal{U} \to \mathbb{R}$ bounded below and some $\varepsilon \geq 0$, we define the strict $\varepsilon$-optimal (relative) sub-level set of $g$,

$$\mathrm{lev}_\varepsilon g = \left\{ u \in \mathcal{U} : g(u) - \inf_{u' \in \mathcal{U}} g(u') < \varepsilon \right\}.$$

Further, we denote $\arg\min g = \arg\min_{u \in \mathcal{U}} g(u)$.

## A.1 Score-and-sort for weak orders ($\mathcal{Z}$)

**Scores and argsort** We consider the case of ranking or selection tasks solved by sorting according to predicted scores. The basic operation we consider, called $\mathrm{argsort}$, is the set-value function that associates to a vector of scores given to items the set of total orders compatible with the scores:

$$\mathrm{argsort} : \mathbb{R}^n \twoheadrightarrow \mathfrak{S}_n \tag{1}$$
$$s \mapsto \left\{ \sigma \in \mathfrak{S}_n : \forall k \in [n-1], s_{\sigma(k)} \geq s_{\sigma(k+1)} \right\}.$$

Note that we consider $\mathrm{argsort}$ as a set-valued function (and not as a function with values in $2^{\mathfrak{S}_n}$), so that follows the usual convention, for any subset $S \in \mathbb{R}^n$, the notation $\mathrm{argsort}(S)$ is defined as:

$$\forall S \subseteq \mathbb{R}^n, \mathrm{argsort}(S) = \bigcup_{s \in S} \mathrm{argsort}(s).$$

**Transpositions** We make heavy use of transpositions: given $i, j$ in $[n]$, the transposition of $i$ and $j$ is the permutation $\tau_{ij}$ such that $\tau_{ij}(i') = i'$ if $i' \notin \{i, j\}$, $\tau_{ij}(j) = i$ and $\tau_{ij}(i) = j$. Given $\sigma$, we denote by $\tau_{ij}\sigma$ the composition of $\tau_{ij}$ and $\sigma$.

Given $z \subseteq \mathfrak{S}_n$, we denote by $\tau_{ij} z = \{\tau_{ij}\sigma : \sigma \in z\}$.

Notice the following difference between a subset $z \subseteq \mathfrak{S}_n$ and a permutation $\sigma \in \mathfrak{S}_n$ when it comes to transpositions: the sequential application of transpositions to a permutation commute and it is an associative operation (because they are all members of the symmetric group). That

is, if $\sigma \in \mathfrak{S}_n$ we have $\tau_{ij}\tau_{i'j'}\sigma = (\tau_{ij}\tau_{i'j'})\sigma = \tau_{i'j'}\tau_{ij}\sigma$. This is not the case in general when applying sequences of transpositions in $\mathcal{Z}$. For instance let $z = \{(123), (213)\}$. Then $\tau_{12}z = z$, so $\tau_{13}\tau_{12}z = \tau_{13}z = \{(321), (231)\}$, while $\tau_{12}\tau_{13}z = \tau_{12}\{(321), (231)\} = \{(312), (132)\}$. Thus, the composition of a permutation and an element of $\mathcal{Z}$ is not defined in general, it is only defined for transpositions, and these should only be read as usual function compositions.

**The decision space**    The final prediction space corresponds to a selection or partial ranking task, generally speaking, a weak order. The decision space $\mathcal{Z}$ should satisfy the following properties:

**(A1).** *The decision space $\mathcal{Z} \subseteq 2^{\mathfrak{S}_n}$ satisfies*

(Total preorders) $\forall z \in \mathcal{Z}, \exists s \in \mathbb{R}^n : z = \mathrm{argsort}(s) \subseteq \mathfrak{S}_n$.

(Partition of $\mathfrak{S}_n$) $\bigcup_{z \in \mathcal{Z}} z = \mathfrak{S}_n$ *and* $\forall z, w \in \mathcal{Z}, (z \neq w) \Rightarrow (z \cap w = \emptyset)$.

(Closed under transpositions) $\forall i, j, \forall z \in \mathcal{Z}, \tau_{ij}z \in \mathcal{Z}$.

Since $\mathcal{Z}$ partitions $\mathfrak{S}_n$, elements of $\mathcal{Z}$ can be seen as equivalence classes on rankings in $\mathfrak{S}_n$, and we can define the function $\Lambda$ as the quotient map which assigns a permutation to its representative member of $\mathcal{Z}$:

$$\Lambda : \mathfrak{S}_n \to \mathcal{Z}$$
$$\sigma \mapsto z \text{ s.t. } \sigma \in z \,,$$

The prediction function is then the composition $\Lambda \circ \mathrm{argsort}$:

$$\forall s \mathbb{R}^n, \ \mathrm{pred}(s) = \Lambda \circ \mathrm{argsort}(s) = \{\Lambda(\sigma) : \sigma \in \mathrm{argsort}(s)\}\,.$$

A straightforward consequence of Assumption (A1) that we make use of when discussing connectedness later is that $\Lambda$ is equivariant by transposition[1]:

$$\forall \sigma, \ \forall i, j, \ \Lambda(\tau_{ij}\sigma) = \tau_{ij}\Lambda(\sigma)\,.$$

**Notation for orders and preorder**    Given $\sigma \in \mathfrak{S}_n$, and $i, j \in [n]$, we use the notation

$$i \succ_\sigma j \ \Leftrightarrow \ \sigma^{-1}(i) < \sigma^{-1}(j) \qquad "\sigma \text{ prefers } i \text{ to } j".$$

This is coherent with an ordering in decreasing order of scores in (1), where $\sigma^{-1}(i)$ is the rank of item $i$ and lower ranks are associated with higher scores (i.e., rank 1 is best), the usual convention in learning to rank.

We use the following notation for the preorder induced by $z$ on $[n]$: $\forall z \in \mathcal{Z}, \forall i, j \in [n]$:

- $i \succeq_z j \ \Leftrightarrow \ \exists \sigma \in z, i \succ_\sigma j,$
- $i \succ_z j \ \Leftrightarrow \ \forall \sigma \in z, i \succ_\sigma j,$
- $\succeq_z$ induces *indifference classes*: $i \sim_z j \ \Leftrightarrow \ \exists \sigma, \sigma' \in z, i \succ_\sigma j$ and $j \succ_{\sigma'} i.$

Given the properties of $\mathcal{Z}$ (Assumption (A1)), the following properties are straightforward to prove: $\forall z \in \mathcal{Z}, \forall s \in \mathbb{R}^n$ s.t. $z = \mathrm{argsort}(s), \forall i, j \in [n]$:

$$a) i \succeq_z j \Leftrightarrow s_i \geq s_j \quad b) i \succ_z j \Leftrightarrow s_i > s_j \quad c) i \sim_z j \Leftrightarrow s_i = s_j \qquad (2)$$
$$d) i \sim_z j \Leftrightarrow \tau_{ij}z = z$$

a), b), c) above simply mean that our definition of $\succeq_z$ is coherent with the preorder induced by the scores. d) is a direct consequence of c). Another noticeable implication of (A1) is that the number of indifference classes is the same for all $z$. This can be seen by combining d) above together with the fact that $z$ is both a partition and stable by transposition.

## A.2 Assumption on task losses

Let $\mathcal{Q}$ be the set of probability mass functions over $\mathcal{Y}$. For any $q \in \mathcal{Q}$, we define the *inner risk* for the task loss $L$ as

$$\forall z \in \mathcal{Z}, \ell(q, z) = \sum_{y \in \mathcal{Y}} q(y) L(y, z) \quad \text{and} \quad \underline{\ell}(q) = \min_{z \in \mathcal{Z}} \ell(q, z)$$

We now detail our assumptions that subsumes the *ranking task loss* definition (Def. 3).

**(A2)** (Items are equivalent *a priori*). $\forall q \in \mathcal{Q}, \forall i, j \in [n], \exists q' \in \mathcal{Q}$ s.t. $\forall z \in \mathcal{Z}, \ell(q, z) = \ell(q', \tau_{ij} z)$.

For the next assumption, for any $z$ and any item $i$, let us denote the rank of $i$ in the preorder $z$ as:

$$z^{-1}(i) = |\{j \in [n] : j \succ_z i\}| + 1.$$

The $+1$ makes sure this definition of rank matches the definition for permutations (first rank is $1$ rather than $0$).

**(A3)** (One distribution over $\mathcal{Y}$ strictly prefers item $i$ over other items). $\forall i \in [n], \exists q_{\text{top}}^{(i)} \in \mathcal{Q}$ such that

1. $\forall z, w \in \mathcal{Z}, \left( z^{-1}(i) < w^{-1}(i) \Rightarrow \ell(q_{\text{top}}^{(i)}, z) < \ell(q_{\text{top}}^{(i)}, w) \right)$

2. $\forall z \in \mathcal{Z}, \forall k \neq i, \forall l \neq i, \ell(q_{\text{top}}^{(i)}, z) = \ell(q_{\text{top}}^{(i)}, \tau_{kl} z)$

The following lemma is a straightforward consequence of the three assumptions above:

**Lemma 14.** *Under assumptions (A1), (A2) and (A3), we have:*

i) $\forall i \in [n], \exists q_{\text{bottom}}^{(i)} \in \mathcal{Q}$ *such that*

    (a) $\forall z, w \in \mathcal{Z}, \left( z^{-1}(i) < w^{-1}(i) \Rightarrow \ell(q_{\text{bottom}}^{(i)}, z) > \ell(q_{\text{bottom}}^{(i)}, w) \right)$

    (b) $\forall z \in \mathcal{Z}, \forall k \neq i, \forall l \neq i, \ell(q_{\text{bottom}}^{(i)}, z) = \ell(q_{\text{bottom}}^{(i)}, \tau_{kl} z)$

ii) $\forall z \in \mathcal{Z}, \exists q^{(z)} \in \mathcal{Q}$ *such that* $\text{argmin}\, \ell(q^{(z)}, .) = \{z\}$.

*Proof.* Notice that without loss of generality, using (A2), we can assume $\forall z, \ell(q_{\text{top}}^{(i)}, z) = \ell(q_{\text{top}}^{(j)}, \tau_{ij} z)$

Then, define

$$q_{\text{bottom}}^{(i)} = \frac{1}{n-1} \sum_{j \neq i} q_{\text{top}}^{(j)}.$$

$q_{\text{bottom}}^{(i)} \in \mathcal{Q}$ by convexity of the simplex. It is then easy to check that $q_{\text{bottom}}^{(i)}$ satisfies the two desired conditions.

For $q^{(z)}$, define $\alpha_i = n - z^{-1}(i)$ and $\tilde{\alpha}_i = \alpha_i / \sum_j \alpha_j$. Then, define

$$q^{(z)} = \sum_i \tilde{\alpha}_i q_{\text{top}}^{(i)}.$$

$q^{(z)} \in \mathcal{Q}$ by convexity of the simplex and it is easy to check that $\text{argmin}\, \ell(q^{(z)}, .) = \{z\}$. $\quad\square$

An immediate consequence of point *ii)* of Lemma 14 above, is the following straightforward result, which we use throughout the proof of the main results to reduce all cases to cases where argmins of the task loss are unique:

**Lemma 15** (Tie-breaking). *Under (A1), (A2) and (A3), $\forall q \in \mathcal{Q}, \forall z \in \text{argmin}\, \ell(q, .), \exists q' \in \mathcal{Q}$ s.t.*

1. $\text{argmin}\, \ell(q', .) = \{z\}$,

2. $\forall z_1, z_2 \in \mathcal{Z}, \ell(q, z_1) > \ell(q, z_2) \Rightarrow \ell(q', z_1) > \ell(q', z_2)$

*Proof.* Let $q \in \mathcal{Q}$ and $z \in \operatorname{argmin} \ell(q,.)$. By point *ii)* of Lemma 14, we know $\exists q^{(z)} \in \mathcal{Q}, \operatorname{argmin} \ell(q^{(z)},.) = \{z\}$. We denote $q_\alpha = (1-\alpha)q + \alpha q^{(z)}$.

1. As $z$ is the only element of $\mathcal{Z}$ optimal for both $q$ and $q^{(z)}$, we have that $\forall \alpha \in (0,1], \operatorname{argmin} \ell(p_\alpha,.) = \{z\}$.

2. Denoting $\epsilon = \min_{z_1,z_2:\ell(q,z_1)\neq\ell(q,z_2)}|\ell(q,z_1) - \ell(q,z_2)|$, Let $0 < \alpha < \frac{\epsilon}{\epsilon+\max_{z'}(\ell(q^{(z)},z')-\underline{\ell}(q^{(z)}))}$. Then, for any $z_1, z_2$ such that $\ell(q,z_1) > \ell(q,z_2)$:

$$\ell(q_\alpha, z_1) - \ell(q_\alpha, z_2) = (1-\alpha)(\ell(q,z_1) - \ell(q,z_2)) + \alpha(\ell(q^{(z)},z_1) - \ell(q^{(z)},z_2))$$
$$\geq (1-\alpha)\epsilon - \alpha\max_{z'\in\mathcal{Z}}(\ell(q^{(z)},z') - \underline{\ell}(q^{(z)})) > 0 \quad \text{for our choice of } \alpha.$$

Thus, for our choice of $\alpha$, $q_\alpha$ satisfies both conditions. $\qquad\qquad\qquad\qquad\square$

### A.3   Notation

For conciseness, in the following, we may skip the sets that universal quantifiers range over when default values apply and are unambiguous:

- $i, j$ always denote items in $[n]$, if the range is not specified, $\forall i, \forall j$ or $\forall i, j$ respectively mean $\forall i \in [n], \forall j \in [n]$ and $\forall i, j \in [n]$
- $k, m$ denote ranks, also in $[n]$. We use the same shorthands as above
- $\mathfrak{S}_n$ denotes the set of permutations of $[n]$. $\sigma, \mu, \sigma'$ denote permutations. $\forall \sigma$ means $\forall \sigma \in \mathfrak{S}_n$, same for $\mu$ and $\sigma'$.
- $z, w, z'$ denote possible predictions, as above, $\forall z$ (without specific range for $z$) means $\forall z \in \mathcal{Z}$
- $q, q'$ are members of $\mathcal{Q}$, i.e., $\forall q$ is a shorthand for $\forall q \in \mathcal{Q}$.
- $s, s'$ are vectors of scores that belong to $\mathbb{R}^n$, $\forall s$ means $\forall s \in \mathbb{R}^n$.

### A.4   Connectedness in $\mathfrak{S}_n$ and $\mathcal{Z}$

**Definition 16** (Connectedness in $\mathfrak{S}_n$). *A subset $\pi \subseteq \mathfrak{S}_n$ is connected in $\mathfrak{S}_n$ if there is a connected subset $S \subseteq \mathbb{R}^n$ such that $\pi = \operatorname{argsort}(S)$.*

**Definition 17** (Connectedness in $\mathcal{Z}$). *A subset $\zeta \subseteq \mathcal{Z}$ is connected in $\mathcal{Z}$ if there is a connected subset $S \subseteq \mathbb{R}^n$ such that $\zeta = \operatorname{pred}(S)$.*

**Remark 18** (Link with topological connectedness). *These notions of connectedness are not* stricto sensu *corresponding to topological connectedness on $\mathfrak{S}_n$ and $\mathcal{Z}$. To derive the topological connectedness that corresponds to these definitions, it is necessary to see* argsort *as a function valued in $2^{\mathfrak{S}_n}$ rather than a set-valued function. When doing so, the image of $\mathbb{R}^n$ by* argsort *is exactly the set of weak orders, on which it is possible to put the specialization topology, where the specialization corresponds to the inclusion relation. It is also sometimes referred to as the Alexandroff topology[2]. This topology makes* argsort *continuous (as a function valued in $2^{\mathfrak{S}_n}$) and open, which explains the use of* argsort *and* pred *to have a suitable (but not topological) definition of connectedness on $\mathfrak{S}_n$ and $\mathcal{Z}$.*

**Definition 19** (u.h.c.). *Let $\mathcal{U}$ and $\mathcal{V}$ be two topological spaces. A set-valued function $g : \mathcal{U} \twoheadrightarrow \mathcal{V}$ is upper hemicontinuous (u.h.c.) if for any non-empty open set $O \subseteq \mathcal{V}$, the upper inverse of $O$ by $g$, $\operatorname{uinv}[g](O) = \{u \in \mathcal{U} : g(u) \subseteq O\}$ is open.*

**Proposition 20.** *Given a discrete space $\mathcal{V}$ (the topology is its whole power set), some $m \in \mathbb{N}$, a set-valued function $g : \mathbb{R}^m \twoheadrightarrow \mathcal{V}$ is u.h.c. if and only if*

$$\forall s \in \mathbb{R}^m, \exists \epsilon > 0, g(\mathcal{B}_\infty(s,\epsilon)) = g(s)$$

*Proof.* Direct implication. Let $s \in \mathbb{R}^n$. Because $\mathcal{V}$ is discrete, $g(s)$ is open. Because $g$ is u.h.c. and $g(s)$ is open, $\operatorname{uinv}[g](g(s))$ is open in $\mathbb{R}^n$. As it contains $s$ itself, $\exists \epsilon > 0, \mathcal{B}_\infty(s,\epsilon) \subseteq \operatorname{uinv}[g](g(s))$. Hence $\exists \epsilon > 0, g(\mathcal{B}_\infty(s,\epsilon)) \subseteq g(s)$. The equality comes from $s$ being itself in the ball.

Reverse implication. Let $O \subseteq \mathcal{V}$ non-empty. For any $s \in \operatorname{uinv}[g](O), \exists \epsilon_s > 0, g(\mathcal{B}_\infty(s,\epsilon_s)) = g(s)$. Thus $\mathcal{B}_\infty(s,\epsilon_s) \subseteq \operatorname{uinv}[g](O)$. Finally, $\operatorname{uinv}[g](O) = \bigcup_{s\in\operatorname{uinv}[g](O)} \mathcal{B}_\infty(s,\epsilon_s)$ which is open. $\square$

Some basic properties of connectedness are given below. These are useful in general to understand basic properties of argsort and pred. Point *vii)* allows to focus on connected subsets $S$ that are open, which is convenient since open subsets or $\mathbb{R}^n$ are connected if and only if they are path-connected, and path-connectedness is easier to manipulate in our case.

**Lemma 21.**

    *i)* $\forall \sigma, \{s : \mathrm{argsort}(s) = \{\sigma\}\}$ *is open and connected.*

    *ii)* $\forall s, \{s' : \mathrm{argsort}(s') = \mathrm{argsort}(s)\}$ *is connected.*

    *iii)* $\forall s, \{s' : \mathrm{argsort}(s') \subseteq \mathrm{argsort}(s)\}$ *is connected.*

    *iv)* $\forall s, \{s' : \mathrm{pred}(s) = \mathrm{pred}(s')\}$ *is connected.*

    *v)* argsort *is u.h.c. with respect to the discrete topology on* $\mathfrak{S}_n$, *i.e.*

$$\forall s, \exists \epsilon > 0, \quad \mathrm{argsort}(\mathcal{B}_\infty(s, \epsilon)) = \mathrm{argsort}(s). \tag{3}$$

    *vi)* $\forall s, \forall \epsilon > 0, \forall \sigma \in \mathrm{argsort}(s), \quad \exists s' : \|s - s'\|_\infty < \epsilon$ *and* $\mathrm{argsort}(s') = \{\sigma\}$.

    *vii)* $\pi \subseteq \mathfrak{S}_n$ *is connected in* $\mathfrak{S}_n$ *if and only if there is an* open *connected subset* $S \subseteq \mathbb{R}^n$ *such that* $\pi = \mathrm{argsort}(S)$. *Similarly,* $\zeta \subseteq \mathcal{Z}$ *is connected if and only if there is an open subset* $S$ *such that* $\zeta = \mathrm{pred}(S)$.

*Proof.*

    i) The set $\{s : \mathrm{argsort}(s) = \{\sigma\}\}$ is open and convex (and thus connected) as the intersection of $n - 1$ open half spaces $\{s_{\sigma(k)} > s_{\sigma(k+1)}\}$ for $k = 1 \ldots n - 1$.

    ii) As for point *i)* above, $\{s' : \mathrm{argsort}(s') = \mathrm{argsort}(s)\}$ is an intersection of open half-spaces and hyperplanes: use the half-space $\{s' : (s_i - s_j)(s'_i - s'_j) > 0\}$ for $i, j$ such that $s_i \neq s_j$ and the hyperplane $\{s' : s'_i = s'_j\}$ for $i, j$ such that $s_i = s_j$. The intersection is convex and thus connected.

    iii) It is a consequence of points *v)* and *vi)*, which are proved below, and point *ii)* above. Fix $s$, denote by $S = \{s' : \mathrm{argsort}(s') \subseteq \mathrm{argsort}(s)\}$ and take an arbitrary $s' \in S$. Denote by $\mathcal{B}_\infty(s, \epsilon) = \{s' : \|s - s'\|_\infty < \epsilon\}$. Fix $\epsilon > 0$ to satisfy (3) for both $s$ and $s'$, i.e., $\mathrm{argsort}(\mathcal{B}_\infty(s, \epsilon)) = \mathrm{argsort}(s)$ and $\mathrm{argsort}(\mathcal{B}_\infty(s', \epsilon)) = \mathrm{argsort}(s')$. let $\sigma \in \mathrm{argsort}(\mathcal{B}_\infty(s', \epsilon)) = \mathrm{argsort}(s')$. Then

$$\mathcal{B}_\infty(s, \epsilon) \cup \mathcal{B}_\infty(s', \epsilon) \cup \{s'' : \mathrm{argsort}(s'') = \{\sigma\}\}$$

is connected and included in $S$. $S$ is thus a union of connected subsets (of the form above) with non-empty intersection $\{s\}$, and is thus connected.

    iv) Fix $s$ and let $s_0$ such that $\mathrm{argsort}(s_0) = \mathrm{pred}(s)$ ($s_0$ exists by definition of pred). The set $\{s' : \mathrm{pred}(s) = \mathrm{pred}(s')\}$ is then equal to $\{s' : \mathrm{argsort}(s) \subseteq \mathrm{argsort}(s_0)\}$, which is connected by point *iii)* above.

    v) To show inclusion, take $\epsilon < \min\{s_i - s_j : i, j \text{ such that } s_i > s_j\}$. The ranking induced by any $s' \in \mathcal{B}_\infty(s, \epsilon)$ is necessarily compatible with a ranking induced by $s$. Equality holds since $s$ is in the open ball.

    vi) Take $\epsilon$ as above, take $s'_i = s_i - \frac{\epsilon \sigma^{-1}(i)}{n}$ (using $-\sigma^{-1}(i)$ because lower ranks are better). by the choice of $\epsilon$, the relative ordering of $s'_i$ and $s'_j$ is the same as $s_i$ and $s_j$ when $s_i \neq s_j$, and the ordering is exactly the one induced by $\sigma$ on equivalence classes of ties in $s$.

    vii) Let $\pi = \mathrm{argsort}(S)$ for some connected $S \subseteq \mathbb{R}^n$. For $s \in S$, let $\epsilon_s > 0$ such that $\mathrm{argsort}(\mathcal{B}_\infty(s, \epsilon_s)) = \mathrm{argsort}(s)$. Then $\pi = \mathrm{argsort}(\bigcup_{s \in S} \mathcal{B}_\infty(s, \epsilon_s))$, and $\bigcup_{s \in S} \mathcal{B}_\infty(s, \epsilon_s)$ is open (as a union of open balls) and connected (as a union of connected sets with a connected set that intersects all of them). The argument is the same for connectedness in $\mathcal{Z}$.

$\square$

The following lemma makes sure the definitions of connectedness in $\mathcal{Z}$ and $\mathfrak{S}_n$ are coherent.

**Lemma 22.** $\zeta \subseteq \mathcal{Z}$ *is connected in* $\mathcal{Z}$ *if and only if* $\bigcup_{z \in \zeta} z$ *is connected in* $\mathfrak{S}_n$.

*Proof.* First point: The if direction holds because $\zeta = \Lambda(\bigcup_{z \in \zeta} z)$, thus if $\bigcup_{z \in \zeta} z = \operatorname{argsort}(S)$ for some connected $S$, then $\zeta = \Lambda \circ \operatorname{argsort}(S) = \operatorname{pred}(S)$. For the only if direction, let us assume $\zeta \subseteq \mathcal{Z}$ is connected, i.e. there is a connected $S \subseteq \mathbb{R}^n$ such that $\zeta = \operatorname{pred}(S)$. Then

$$\bigcup_{z \in \zeta} z = \bigcup_{s \in S} \operatorname{pred}(s) = \bigcup_{s \in S} \operatorname{argsort}(\{s' : \operatorname{argsort}(s') \subseteq \operatorname{pred}(s)\})$$

$$= \operatorname{argsort}(\underbrace{\bigcup_{s \in S}\{s' : \operatorname{argsort}(s') \subseteq \operatorname{pred}(s)\}}_{\tilde{S}})$$

Thee set $\tilde{S}$ is the union of connected sets (by Lemma 21 point *iii)*), and there is a connected set (S) that intersects all members of the union. Thus, it is connected. $\square$

**Remark 23.** *An analog statement from* $\mathfrak{S}_n$ *to* $\mathcal{Z}$ *is false in general: there exists* $n$ *and* $\operatorname{pred}$ *satisfying all assumptions such that there is* $\pi \subseteq \mathfrak{S}_n$ *not connected but* $\Lambda(\pi)$ *is connected.*

*To see this, we anticipate a bit on the idea of paths of adjacent items (Lemma 36 below). Consider a top-2 selection task with n=3. Let* $\pi = \{2 \succ 1 \succ 3, 3 \succ 1 \succ 2\}$. $\pi$ *is not connected because there is no path between the two permutations. However,* $\Lambda(\pi) = \{1 \sim 2 \succ 3, 1 \sim 3 \succ 2\}$, *which is connected because there is a path (transposition of* 2 *and* 3*).*

# B  Calibration and Consistency

*Disclaimer: As mentioned in the body of the paper, we address, in these Appendices, a setting more general than ranking. This setting is described in Section A. We restate first the statement of the paper (on ranking), then state the more general statement (on weak orders) and prove the latter.*

**Subsection B.1**  describes the definition and basic notions of calibration and uniform calibration,

**Subsection B.2**  provides the proof that calibration is not just an inclusion of argmins, but rather an equality, stated in Theorem 4 (also [7, Th. 2]),

**Subsection B.3**  contains the proof that uniform calibration is equivalent calibration under our assumptions as mentioned in Proposition 2.

## B.1  Calibration and Consistency

Following the body of the paper, a surrogate loss $\Phi : \mathcal{Y} \times \mathbb{R}^n \to \mathbb{R}_+$ is a measurable function that aims to be minimized. Similarly to the task loss, we can define the outer and inner risks, given a function $f : \mathcal{X} \to \mathbb{R}^n$ belonging to the space of measurable function from $\mathcal{X}$ to $\mathbb{R}^n$, denoted $\mathcal{F}$, a distribution $P$ over $\mathcal{X} \times \mathcal{Y}$, $s \in \mathbb{R}^n$ and a distribution $q \in \mathcal{Q}$,

$$\mathcal{R}_{\Phi,P}(f) = \mathbb{E}_P[\Phi(Y, f(X))], \qquad \phi(q, s) = \mathbb{E}_q[\Phi(Y, s)], \qquad \underline{\phi}(q) = \inf_{s' \in \mathbb{R}^n} \phi(q, s').$$

We can state the definition of calibration in $\mathcal{Z}$,

**Definition 24** (calibration). $\Phi$ *is L-calibrated if and only if*

$$\forall q \in \mathcal{Q}, \exists \delta > 0, \ \operatorname{pred}(\operatorname{lev}_\delta \phi(q, .)) \subseteq \operatorname{argmin} \ell(q, .)$$

And the one of uniform calibration,

**Definition 25** (uniform calibration). $\Phi$ *is L-uniformly calibrated if and only if*

$$\forall \varepsilon > 0, \exists \delta > 0, \forall q \in \mathcal{Q}, \ \operatorname{pred}(\operatorname{lev}_\delta \phi(q, .)) \subseteq \operatorname{lev}_\varepsilon \ell(q, .)$$

Indeed, uniform calibration leads to the existence of an excess risk bound allowing to control the task risk by the surrogate risk.

**Definition 26** (excess risk bound). *A continuous function $\delta : \mathbb{R}_+ \to \mathbb{R}_+$ is an* excess risk bound *if*

$$\forall P, \forall f \in \mathcal{F}, \quad \mathcal{R}_{L,P}(\text{pred} \circ f) - \inf_{h \in \mathcal{H}} \mathcal{R}_{L,P}(h) \leq \delta \left( \mathcal{R}_{\Phi,P}(f) - \inf_{g \in \mathcal{F}} \mathcal{R}_{\Phi,P}(g) \right)$$

*and*

$$\delta(\epsilon) \xrightarrow[\epsilon \to 0]{} 0$$

**Proposition 27.** $\Phi$ *is uniformly L-calibrated if and only if there exists an excess risk bound.*

*Proof.* This result is directly derived from Steinwart [31] Th. 2.13 (direct implication) and Th. 2.17 (reverse implication). □

## B.2 Proof of characterization of Calibration in Theorem 4 (equality of optimal predictions)

In this section, for the sake of completeness, we provide a full proof for the generalization of Theorem 4 under our assumptions. First, lets remind the exact statement of the paper in the ranking setting.

**Theorem 4** ([7, Th. 2]). *Given a ranking loss L, $\Phi : \mathcal{Y} \times \mathbb{R}^n \to \mathbb{R}_+$ is L-calibrated if and only if*

$$\forall q \in \mathcal{Q}, \exists \delta_0 > 0, \forall \delta \in (0, \delta_0], \quad \underset{\sigma \in \mathfrak{S}_n}{\operatorname{argmin}} \ell(q, \sigma) = \bigcup_{s:\phi(q,s) - \underline{\phi}(q) < \delta} \operatorname{argsort}(s)$$

Then, the version of the same theorem in $\mathcal{Z}$,

**Theorem 28.** *Assuming (A1), (A2) and (A3), $\Phi$ is L-calibrated if and only if*

$$\forall q \in \mathcal{Q}, \exists \delta_0 > 0, \forall 0 < \delta < \delta_0, \operatorname{pred}(\operatorname{lev}_\delta \phi(q, .)) = \operatorname{argmin} \ell(q, .)$$

Before providing to the proof of Theorem 28, we state a general result of upper hemicontinuity that underlies many of the proofs in these appendices. It is a simplified version of the Berge Maximum Theorem [24]. One application is that $\operatorname{argmin} \ell(q, .)$ is u.h.c. as a function of $q$.

**Lemma 29.** *Given a function $g : \mathcal{Q} \times \mathcal{Z} \to \mathbb{R}$, continuous in its first argument, the set-valued function*

$$\begin{cases} \mathcal{Q} & \twoheadrightarrow \mathcal{Z} \\ q & \mapsto \underset{z \in \mathcal{Z}}{\operatorname{argmin}} \, g(q, z) \end{cases} \qquad \text{is u.h.c.}$$

*Proof.* We use the characterization of u.h.c. from Prop.20 as $\mathcal{Q} \subseteq \mathbb{R}^{|\mathcal{Y}|}$. Let $q \in \mathcal{Q}$ be a distribution. We denote $\mathcal{A}_q = \operatorname{argmin}_{z \in \mathcal{Z}} g(q, z)$ and

$$\epsilon_q = \min_{z \in \mathcal{A}_q, w \in \mathcal{Z} \setminus \mathcal{A}_q} |g(q, z) - g(q, w)|$$

For some $\delta > 0$, and for any $\tilde{q} \in \mathcal{B}_\infty(q, \delta)$, we use the following decomposition,

$$g(\tilde{q}, w) - g(\tilde{q}, z) = \underbrace{g(\tilde{q}, w) - g(q, w)}_{\substack{\exists \delta_1 > 0, \ \_ < \frac{\epsilon_q}{4} \\ \text{(continuity)}}} + \underbrace{g(q, w) - g(q, z)}_{> \epsilon_q} + \underbrace{g(q, z) - g(\tilde{q}, z)}_{\substack{\exists \delta_2 > 0, \ \_ < \frac{\epsilon_q}{4} \\ \text{(continuity)}}}$$

Thanks to the continuity of $g$ in its first argument, we have that

$$\exists \delta_q > 0, \forall \tilde{q} \in \mathcal{B}_\infty(q, \delta), g(\tilde{q}, w) - g(\tilde{q}, z) > \frac{\epsilon_q}{2}.$$

Thus,

$$\exists \delta_q > 0, \bigcup_{\tilde{q} \in \mathcal{B}_\infty(q, \delta)} \underset{z \in \mathcal{Z}}{\operatorname{argmin}} \, g(\tilde{q}, z) \subseteq \underset{z \in \mathcal{Z}}{\operatorname{argmin}} \, g(q, z)$$

The equality comes from the fact $q$ itself is in the ball. □

We can now proceed to the proof of Theorem 28.

*Proof of Theorem 28.*

$(\Phi, \mathrm{pred})$ is $L$-calibrated

$$\Leftrightarrow \forall q \in \mathcal{Q}, \exists \delta_0 > 0, \forall 0 < \delta < \delta_0, \mathrm{pred}(\mathrm{lev}_\delta \phi(q, .)) \subseteq \mathrm{argmin}\, \ell(q, .) \quad \text{(Definition 25)}$$

We now focus on showing

$$\forall q \in \mathcal{Q}, \exists \delta_0 > 0, \forall 0 < \delta < \delta_0, \mathrm{pred}(\mathrm{lev}_\delta \phi(q, .)) \subseteq \mathrm{argmin}\, \ell(q, .)$$
$$\Leftrightarrow$$
$$\forall q \in \mathcal{Q}, \exists \delta_0 > 0, \forall 0 < \delta < \delta_0, \mathrm{pred}(\mathrm{lev}_\delta \phi(q, .)) = \mathrm{argmin}\, \ell(q, .)$$

Or otherwise stated, that

$$\forall q \in \mathcal{Q}, \bigcap_{\delta > 0} \mathrm{pred}(\mathrm{lev}_\delta \phi(q, .)) \subseteq \mathrm{argmin}\, \ell(q, .) \ \Leftrightarrow \ \forall q \in \mathcal{Q}, \bigcap_{\delta > 0} \mathrm{pred}(\mathrm{lev}_\delta \phi(q, .)) = \mathrm{argmin}\, \ell(q, .)$$
$$(4)$$

First, for any $q \in \mathcal{Q}$, after defining $\tilde{\phi}(q, z) = \inf_{s \in \mathrm{pred}^{-1}(z)} \phi(q, s)$, we have that

$$\bigcap_{\delta > 0} \mathrm{pred}(\mathrm{lev}_\delta \phi(q, .)) = \mathrm{argmin}\, \tilde{\phi}(q, .).$$

Now, thanks to Wijsman [37, Th. 2], we know that $\tilde{\phi}$ is continuous with respect to its first argument, then Lemma 29 gives us that $\mathrm{argmin}\, \tilde{\phi}(q, .)$ is u.h.c. and thanks to Prop. 20, we have

$$\exists \delta > 0, \bigcup_{\tilde{q} \in B_\delta(q)} \mathrm{argmin}\, \tilde{\phi}(\tilde{q}, .) = \mathrm{argmin}\, \tilde{\phi}(q, .) = \bigcap_{\delta > 0} \mathrm{pred}(\mathrm{lev}_\delta \phi(q, .)). \tag{5}$$

Let's take $z \in \mathrm{argmin}\, \ell(q, .)$. By Lemma 14 *ii)*, $\exists q^z \in \mathcal{Q}, \mathrm{argmin}\, \ell(q^z, .) = \{z\}$ and we can define $\tilde{q}^\alpha = (1 - \alpha)q + \alpha q^z$ for some $\alpha \in (0, 1)$. Because $\mathrm{argmin}\, \ell(q, .) \cap \mathrm{argmin}\, \ell(q^z, .) = \{z\}$, then $\mathrm{argmin}\, \ell(\tilde{q}^\alpha, .) = \{z\}$ ($z$ is the only element optimal for both components of the mixture). Finally, we prove (4):

$$\forall q \in \mathcal{Q}, \bigcap_{\delta > 0} \mathrm{pred}(\mathrm{lev}_\delta \phi(q, .)) \subseteq \mathrm{argmin}\, \ell(q, .)$$

$$\Rightarrow \forall q \in \mathcal{Q}, \forall z \in \mathrm{argmin}\, \ell(q, .), \forall \alpha \in (0, 1), \bigcap_{\delta > 0} \mathrm{pred}(\mathrm{lev}_\delta \phi(\tilde{q}^\alpha, .)) = \{z\} \qquad \text{(previous line applied at } \tilde{q}^\alpha)$$

$$\Rightarrow \forall q \in \mathcal{Q}, \forall z \in \mathrm{argmin}\, \ell(q, .), \forall \alpha \in (0, 1), \mathrm{argmin}\, \tilde{\phi}(\tilde{q}^\alpha, .) = \{z\} \qquad \text{(from (5))}$$

$$\Rightarrow \forall q \in \mathcal{Q}, \forall z \in \mathrm{argmin}\, \ell(q, .), \exists \alpha \in (0, 1), \{z\} = \mathrm{argmin}\, \tilde{\phi}(\tilde{q}^\alpha, .) \subseteq \mathrm{argmin}\, \tilde{\phi}(q, .) \qquad \text{(from (5))}$$

$$\Rightarrow \forall q \in \mathcal{Q}, \forall z \in \mathrm{argmin}\, \ell(q, .), \{z\} \subseteq \bigcap_{\delta > 0} \mathrm{pred}(\mathrm{lev}_\delta \phi(q, .)) \qquad \text{(from (5) again)}$$

$$\Rightarrow \forall q \in \mathcal{Q}, \mathrm{argmin}\, \ell(q, .) \subseteq \bigcap_{\delta > 0} \mathrm{pred}(\mathrm{lev}_\delta \phi(q, .))$$

$\square$

## B.3 Equivalence between calibration and uniform calibration

In this section, we prove the equivalence between calibration and the existence of an excess risk bound stated in Proposition 2. First, we remind the exact statement of the paper in the ranking setting.

**Proposition 2.** $\Phi$ *is $L$-calibrated if and only if there is an excess risk bound between $\mathcal{R}_\Phi$ and $\mathcal{R}_L$.*

Since we already established in Proposition 27 that uniform calibration is equivalent to the existence of an excess risk bound, we now show the equivalence between calibration and uniform calibration in our general setting:

**Theorem 30.** *Assuming (A1), $\Phi$ is $L$-calibrated if and only if $\Phi$ is $L$-uniformly calibrated.*

*Proof.* If $\Phi$ is $L$-uniformly calibrated, then it is $L$-calibrated, so we just have to prove the only if direction.

Using the notation $\tilde{\phi}(q, z) = \inf_{s \in \mathrm{pred}^{-1}(z)} \phi(q, s) \in \mathbb{R}$ for $z \in \mathcal{Z}$, let us notice that all the following functions are continuous on $\mathcal{Q}$ as $\mathcal{Y}$ is finite [37, Theorem 2]:

$$q \mapsto \tilde{\phi}(q, z) \qquad\qquad q \mapsto \underline{\phi}(q) = \inf_{s \in \mathbb{R}^n} \phi(q, s) = \min_{z \in \mathcal{Z}} \tilde{\phi}(q, z)$$

$$q \mapsto \ell(q, z) \qquad\qquad q \mapsto \underline{\ell}(q) = \min_{z \in \mathcal{Z}} \ell(q, z)\,.$$

Consequently, for every $z \in \mathcal{Z}$, both functions

$$g_z : q \mapsto \ell(q, z) - \underline{\ell}(q) \text{ and} \qquad\qquad h_z : q \mapsto \tilde{\phi}(q, z) - \underline{\phi}(q)$$

are continuous on $\mathcal{Q}$. Also notice that since both $\mathcal{Y}$ and $\mathcal{Z}$ are finite, both $\ell$ and $\tilde{\phi}$ are bounded on their domain. More formally, denoting $B_\ell = \max_{z \in \mathcal{Z}} \max_{y \in \mathcal{Y}} L(y, z) - \min_{z \in \mathcal{Z}} \min_{y \in \mathcal{Y}} L(y, z)$, we have $g_z \in [0, B_\ell]$.

Let us now fix $\epsilon > 0$ and denote

$$\tilde{\Delta}(z, \epsilon) = \left\{ q \in \mathcal{Q} : g_z(q) \geq \epsilon \right\} = g_z^{-1}([\epsilon, B_\ell])\,.$$

Since $g_z$ is continuous from a compact set ($\mathcal{Q}$) to $\mathbb{R}$, it is a proper map, which means that preimages of compact subsets of $\mathbb{R}$ are compact. In particular, it implies that $\tilde{\Delta}(z, \epsilon)$ is compact. Since $h_z$ is also continuous on $\mathcal{Q}$, it implies that is reaches its minimum on $\tilde{\Delta}(z, \epsilon)$. Let us then denote:

$$\delta(\epsilon) = \min_{z \in \mathcal{Z}} \min_{q \in \tilde{\Delta}(z, \epsilon)} h_z(q)\,.$$

Then, since $\Phi$ is $L$-calibrated, $g_z(q) > \epsilon \Rightarrow h_z(q) > 0$ for all $z, q$ and thus $\delta(\epsilon) > 0$.

Thus, for all $\epsilon > 0$, $q \in \mathcal{Q}$ and $s \in \mathbb{R}^n$, we have:

$$\phi(q, s) - \underline{\phi}(q) < \delta(\epsilon)$$
$$\Rightarrow \qquad \forall z \in \mathrm{pred}(s), \quad \tilde{\phi}(q, z) - \underline{\phi}(q) < \delta(\epsilon) \qquad\qquad \text{(by def. of } \tilde{\phi})$$
$$\Rightarrow \qquad \forall z \in \mathrm{pred}(s), \quad \ell(q, z) - \underline{\ell}(q) < \epsilon \qquad\qquad \text{(by Def of } \delta(\epsilon))$$
$$\Rightarrow \qquad \ell(q, s) - \underline{\ell}(q) < \epsilon$$

which means that $\Phi$ is $L$-uniformly calibrated. $\qquad\qquad\square$

## C  Proof of the main result

*Disclaimer: As mentioned in the body of the paper, we address, in these Appendices, a setting more general than ranking. This setting is described in Section A. We restate first the statement of the paper (on ranking), then state the more general statement (on weak orders) and prove the latter.*

The objective of this section is to prove the following Theorem 6. First we restate the theorem in the ranking setting.

**Theorem 6.** *For a ranking loss $L$, the following statements are equivalent:*

*(i)  $L$ is CEU,*

*(ii)  $\forall q, \mathrm{argmin}_\sigma \ell(q, \sigma)$ is connected.*

*(iii)  $\forall \epsilon > 0, \forall q, \mathrm{lev}_\epsilon \ell(q, .)$ is connected,*

*Moreover, the function $\tilde{u} : \mathcal{Y} \to \mathbb{R}^n$ defined as:* $\quad \forall i \in [n], \tilde{u}_i(y) = -\sum\limits_{\sigma \in \mathfrak{S}_n} \mathbb{1}_{\{\sigma(1) = i\}} L(y, \sigma)$

*is a utility function for $L$ whenever there exists a utility function for $L$ (i.e., whenever $L$ is CEU).*

Before stating its generalization in $\mathcal{Z}$, we need to note the definition of CEU task loss extends in a strait-forward way to $\mathcal{Z}$. We state it for the sake of completeness.

**Definition 31** (CEU). *A task loss $L$ is* compatible with expected utility *(CEU) if there exists a function $u : \mathcal{Y} \to \mathbb{R}^n$ such that $\Phi_u^{\mathrm{sq}} : y, s \mapsto (u(y) - s)^2$ is $L$-calibrated.*

**Theorem 32.** *Under assumptions (A1), (A2) and (A3), for a task loss L, the following statements are equivalent:*

(i) *L is CEU,*

(ii) $\forall q, \operatorname{argmin} \ell(q, .)$ *is connected.*

(iii) $\forall \epsilon > 0, \forall q, \operatorname{lev}_\epsilon \ell(q, .)$ *is connected,*

*Moreover, the function $\tilde{u} : \mathcal{Y} \to \mathbb{R}^n$ defined as:* $\quad \forall i \in [n], \tilde{u}_i(y) = - \displaystyle\sum_{z \in \mathcal{Z} : z^{-1}(i)=1} L(y, z)$

*is a utility function for L whenever there exists a utility function for L (i.e., whenever L is CEU).*

We now proceed with the proof that this whole section is dedicated to. The two main intermediary results of this proof are Corollary 41 and Theorem 42.

In Appendix C.1, we elaborate on more properties of connectedness in $\mathfrak{S}_n$ and $\mathcal{Z}$. In particular, we prove Lemma 36 which is the generalized version of Proposition 9 in the main paper regarding the characterization of connectedness with paths of adjacent items. This proposition is a key technical result that is fundamental to the results of Appendices D, E and H.

In Appendix C.2, we give technical lemmas that lead to the main steps in the full proof. The main result of this section, Lemma 38 allows to define sequences of distributions in $cQ$ for which the argmin matches a path of adjacent transposition between permutation. This is the key component to extend properties of argmins to properties of the entire loss.

In Appendix C.3, we put the tools together, and first prove the strict monotonicity of loss with connected argmins (Corollary 41), which means that preferences between items as defined by optimal $z$ are reflected by preferences even for non-optimal $z'$. This, in turn, allow us to prove the existence of the utility function and its analytical formula 42. The final proof at the end of that section makes the link with connected sublevel sets.

## C.1 Properties of connected sets in $\mathfrak{S}_n$ and $\mathcal{Z}$

The main technical property of connectedness is the existence of paths with transpositions of adjacent items:

**Definition 33** (Adjacent items). $\forall \sigma \in \mathfrak{S}_n$, *items $i$ and $j$ are* adjacent[2] *in $\sigma$ if $|\sigma^{-1}(i) - \sigma^{-1}(j)| \leq 1$.*

$\forall z \in \mathcal{Z}$, *items $i$ and $j$ are* adjacent *in $z$ if $\exists \sigma \in z$ such that $i$ and $j$ are adjacent in $\sigma$.*

**Definition 34** (paths of adjacent items).
**Path in $\mathfrak{S}_n$:** *Let $\pi \subseteq \mathfrak{S}_n$, and $\sigma, \sigma' \in \pi$. A path of adjacent transpositions of length $M \in \mathbb{N}$ from $\sigma$ to $\sigma'$ in $\pi$ is a sequence $(\sigma_0, ..., \sigma_M) \in \pi^M$ such that:*

- $\sigma_0 = \sigma$ *and* $\sigma_M = \sigma'$

- $\forall m \in [M]$, *there is $(i_m, i'_m) \in [n]^2$ such that $i_m$ and $i'_m$ are adjacent in $\sigma_{m-1}$ and $\sigma_m = \tau_{i_m i'_m} \sigma_{m-1}$.*

**Path in $\mathcal{Z}$:** *Let $\zeta \subseteq \mathcal{Z}$, and $z, z' \in \zeta$. A path of adjacent transpositions of length $M \in \mathbb{N}$ from $z$ to $z'$ in $\zeta$ is a sequence $(z_0, ..., z_M) \in \zeta^M$ such that:*

- $z_0 = z$ *and* $z_M = z'$

- $\forall m \in [M]$, *there is $(i_m, i'_m) \in [n]^2$ such that $i_m$ and $i'_m$ are adjacent in $z_{m-1}$ and $z_m = \tau_{i_m i'_m} z_{m-1}$.*

We say that there is a path from $\sigma$ and $\sigma'$ in $\pi \subseteq \mathfrak{S}_n$ if there is $M \in \mathbb{N}$ such that there is a path of adjacent transpositions of length M from $\sigma$ to $\sigma'$ in $\pi$. A similar terminology is used in $\mathcal{Z}$.

Paths of adjacent items between two permutations can be found by bubble sort. We describe a naive version of the algorithm and then explain the important property that we use later in the proofs.

---

**Algorithm 1:** (Naive) Bubble Sort

---

**input** : $n$ and $\sigma, \sigma' \in \mathfrak{S}_n$
**output** : a sequence of transpositions $\tau^{(1)}, \dots, \tau^{(t)}$ such that $\sigma = \tau^{(t)} \dots \tau^{(1)} \sigma'$

$k = n$;
$\nu = \sigma'$;
$t = 1$;
**for** $m \leftarrow 1$ **to** $n - 1$ **do**
    **for** $k \leftarrow 1$ **to** $m - i$ **do**
        **if** $\sigma^{-1}(\nu(k)) > \sigma^{-1}(\nu(k+1))$ **then**
            $\tau^{(t)} = \tau_{\nu(k)\nu(k+1)}$;
            $\nu = \tau^{(t)}\nu$;
            $t = t + 1$;
        **end**
    **end**
**end**

---

**Remark 35.** *Note the* monotonicity *property of the bubble sort. We use it later in the proofs. Given two items $i \neq j \in [n]$, if $i \succ_\sigma j$ and $i \prec_{\sigma'} j$, then either $\tau_{ij}$ or $\tau_{ji}$ appears exactly once in the output sequence $\tau^{(1)}, \dots, \tau^{(t)}$ of bubble sort. Otherwise, neither $\tau_{ij}$ nor $\tau_{ji}$ appears in the output sequence of bubble sort.*

The next lemma is the general version of Proposition 9 in the main paper.

**Lemma 36** (connectedness and paths in $\mathfrak{S}_n$). *$\pi \subseteq \mathfrak{S}_n$ is connected in $\mathfrak{S}_n$ if and only if for every $\sigma, \sigma' \in \pi$, there is a path from $\sigma$ to $\sigma'$ in $\pi$.*

*As a corollary, $\zeta \subseteq \mathcal{Z}$ is connected in $\mathcal{Z}$ if and only if for every $z, z' \in \zeta$ there is a path between $z$ and $z'$ in $\zeta$.*

*Proof.* **"if" direction** Let $\sigma$ and $\sigma'$ two permutations such that there is a path between them in $\pi$. Denote $(\sigma_m)_{m=0}^M$ the sequence of permutations along that path ($\sigma_0 = \sigma$ and $\sigma_M = \sigma'$). We show that there is a connected set $S$ such that $\{\sigma_0, \dots, \sigma_M\} = \mathrm{argsort}(S)$.

The first step is to notice that if $\sigma_m$ and $\sigma_{m+1}$ are two permutations that are equal up to an adjacent transposition, then there is $s^{(m)}$ such that $\mathrm{argsort}(s^{(m)}) = \{\sigma_m, \sigma_{m+1}\}$ (use $s_i^{(m)} = -(\sigma_m^{-1}(i) + \sigma_{m+1}^{-1}(i))$). We thus have:

$$\{\sigma_0, \dots, \sigma_M\} = \mathrm{argsort}\left( \bigcup_{m=0..M-1} \underbrace{\{s : \mathrm{argsort}(s) \subseteq \mathrm{argsort}(s^{(m)})\}}_{S_m} \right)$$

Each $S_m$ is connected by Lemma 21 point *iii)*, and $S_m \cap S_{m+1} = \{s : \mathrm{argsort}(s) = \sigma_{m+1}\}$ is open and connected. Thus $\bigcup_m S_m$ is connected. Finally, $\pi$ is the union of all these sets for all possible pairs $\sigma, \sigma' \in \pi$. The union itself is connected

**"only if" direction** We prove the result in two steps. First, we prove that if $\pi = \mathrm{argsort}(s)$ for some $s \in \mathbb{R}^n$, then any two $\sigma, \sigma'$ are connected by a path. We then go to the more general case $\pi = \mathrm{argsort}(S)$ with $S \subseteq \mathbb{R}^n$.

*Case 1: $\exists s \in \mathbb{R}^n, \pi = \mathrm{argsort}(s)$.* Let $\sigma, \sigma' \in \pi$. By definition of argsort, $i \succ_\sigma j$ and $j \succ_{\sigma'} i$ is only possible for $i, j$ such that $s_i = s_j$. Apply `bubblesort(target=`$\sigma$`, input=`$\sigma'$`)`. Bubble sort gives a path $(\sigma_m)_{m=0}^M$ between $\sigma$ and $\sigma'$; applying recursively the above remark to $i \succ_\sigma j$ and $j \succ_{\sigma_m} i$, the adjacent transpositions only exchange items $i, j$ that are tied in $s$. Thus, by the definition of argsort, $\forall m \in \{0, \dots, M\}, \sigma_m \in \mathrm{argsort}(s) = \pi$.

*Step 2: general case: $\pi = \mathrm{argsort}(S)$ for connected $S \subseteq \mathbb{R}^n$* Using Lemma 21 (point *vii)*) we can assume without loss of generality that $S$ is open and thus path-connected. Let $\sigma, \sigma' \in \pi$. Let $\gamma : [0,1] \to S$ be a continuous function such that $\sigma \in \mathrm{argsort}(\gamma(0))$ and $\sigma' \in \mathrm{argsort}(\gamma(1))$; such a $\gamma$ exists by the assumption $\pi = \mathrm{argsort}(S)$ for path-connected $S$. By definition of $\gamma$, we have

$\text{argsort}(\gamma([0,1])) \subseteq \pi$. Now consider the undirected graph $G = (\pi, E)$ with set of nodes $\pi$ and edges $E$, where $(\nu, \nu') \in E$ if $\exists t \in [0,1]$ such that $\{\nu, \nu'\} \subseteq \text{argsort}(\gamma(t))$. We now prove that there is a path in $G$ (in the usual sense of paths in a graph) between the nodes corresponding to $\sigma$ and $\sigma'$. To that end, first notice that by Lemma 21 point *v)*, $\text{argsort}(\gamma(t - \epsilon)) \subseteq \text{argsort}(\gamma(t))$ for small enough $\epsilon$ and $t \in (0,1]$, since $\gamma$ is continuous (a similar statements holds for $t + \epsilon$ instead of $t - \epsilon$). This means that all permutations in consecutive values of $t \mapsto \text{argsort}(\gamma(t))$ are neighbors in $G$. Now let us consider

$$t_0 = \sup\{t \in [0,1] : \forall \nu \in \text{argsort}(\gamma(t)), \nu \text{ is connected to } \sigma \text{ in } G\}.$$

$t_0$ is well defined since all permutations of $\gamma(0)$ are connected to $\sigma$ in $G$. First notice that since argsort is u.h.c. (Lemma 21 point *v)*), the sup above is actually a max. We now prove that it implies that all permutations in $\text{argsort}(\gamma(t))$ are connected to $\sigma$ in $G$. Aiming for a contradiction, assume $t_0 < 1$. Then, by the remark above, there is $\epsilon > 0$ such that $\text{argsort}(\gamma(t_0 + \epsilon)) \subseteq \text{argsort}(\gamma(t_0))$. Then, all permutations of $\text{argsort}(\gamma(t_0 + \epsilon))$ are connected in $G$ to all permutations of $\text{argsort}(\gamma(t_0))$, and are thus connected to $\sigma$ in $G$. This contradicts the definition of $t_0$.

Thus, all permutations in $\gamma(1)$ are connected in $G$ to all permutations in $\gamma(0)$. To finish the proof, let us remind that by the Step 1 above, for all $t$, all permutations within $\text{argsort}(\gamma(t))$ are connected by a path (of adjacent transpositions) in $\mathfrak{S}_n$. Thus, any two permutations connected in $G$ are also connected by a path of adjacent transpositions. Thus, $\sigma$ and $\sigma'$ are connected by a path of adjacent transpositions. $\qquad\square$

## C.2  Main property of Losses with connected argmins

To clarify the results of the section, we state the connectedness of the argmins of the losses as an assumption:

**(A4).** $\forall q \in \mathcal{Q}, \text{argmin}\, \ell(q,.)$ *is connected in* $\mathcal{Z}$.

The proof of the final result is the combination of the following observation, which is a general property of preorders, together with the lemma that follows.

The general observation is the following: Given two preorders $z$ and $w$ that have the same (strict) preferences between two items $i$ and $j$, then we can find a path of adjacent transpositions between them that never transposes $i$ and $j$. Thus the strict preferences between $i$ and $j$ are kept constant along the path.

**Lemma 37.** $\forall z, w \in \mathcal{Z}, \forall i, j \in [n]$ *s.t.* $i \succ_z j$ *and* $i \succ_w j$, $\exists M \in \mathbb{N}, \exists (i_m, i'_m)_{m=0}^M \in [n]^{2M}$ *s.t. denoting* $z_0 = z$ *and* $z_m = \tau_{i_m, i'_m} z_{m-1}$, *we have*

   *(i)* $\forall m \in [M], i_q, i'_m$ *are adjacent in* $z_{m-1}$,

   *(ii)* $z_M = w$,

   *(iii)* $\forall m \in [M], \{k_m, k'_m\} \neq \{i, j\}$.

*Proof.* For two permutations, the existence of such a path is given by bubble sort: since bubble sort is monotonic (see Remark 35), it never exchanges two items that are in an ordering compatible with the target weak order. The result in $\mathcal{Z}$ follows by compositing with $\Lambda$ the path between $\sigma \in z$ and $\sigma' \in z'$. $\qquad\square$

The next lemma is the main technical step in the proof of our final result, and transposes the previous lemma in sequences of argmins for different distributions, when the argmins are always connected. In a similar way to the path between preorders above, the lemma considers a distribution for which the argmin of the loss is a single preorder $z$, and two fixed items $i, j$. Then, it shows that for any two adjacent items $\{k, l\} \in z$, different from $\{i, j\}$, we can find a distribution for which the argmin is $\tau_{kl} z$, which at the same time preserves strict preferences (by the inner risk) between $z'$ and $\tau_{ij} z'$:

**Lemma 38** (Adjacent transpositions of argmin). *Under assumptions (A1), (A2) and (A3) and (A4):* $\forall q, z$ *s.t.* $\text{argmin}\, \ell(q,.) = \{z\}, \forall i, j, k, l \in [n]$ *s.t.* $\{i, j\} \neq \{k, l\}$ *and* $(k, l)$ *are adjacent in* $z$, *we have:* $\exists q' \in \mathcal{Q}$ *s.t.*

   *1.* $\text{argmin}\, \ell(q',.) = \{\tau_{kl} z\}$,

2. $\forall z', \ell(q, z') > \ell(q, \tau_{ij} z') \Rightarrow \ell(q', z') > \ell(q', \tau_{ij} z')$

*Proof.* Note that the result trivially holds with $q' = q$ when $k \sim_z l$, since in that case $\tau_{kl} z = z$ (by (2) d)). We thus focus on the case $k \not\sim_z l$. Without loss of generality, we assume $l \succ_z k$.

**First case: $k \neq i$ and $k \neq j$.** Consider $q_\alpha = (1 - \alpha)q + \alpha q_{\text{top}}^{(k)}$ and let $\alpha_0 = \inf\{\alpha : \{z\} \neq \arg\min \ell(q_\alpha, .)\}$. Notice that $\alpha_0$ is well defined since all $z' \in \arg\min \ell(q_1, .)$ must have $z'^{-1}(k) = 1$ by definition of $q_{\text{top}}^{(k)}$, while $z^{-1}(k) > z^{-1}(l)$ by our assumption $l \succ_z k$. Moreover, since $\arg\min \ell(., .)$ is uhc, the infimum is a minimum, and thus $\alpha_0 > 0$ since $\arg\min \ell(q_0, .) = \{z\}$.

Let $q' = q_{\alpha_0}$. Since $\arg\min \ell(., .)$ is uhc, we have that $\exists \epsilon > 0$ such that $\arg\min \ell(\mathcal{B}_\infty(q', \epsilon), .) = \arg\min \ell(q', .)$. Taking one such $\epsilon$ with the additional requirement $\epsilon < \alpha_0$, we have:

1. $z \in \arg\min \ell(q', .)$, because $\{z\} = \arg\min \ell(q_{\alpha_0 - \epsilon}, .) \subseteq \arg\min \ell(q', .)$. The first equality comes from the definition of $\alpha_0$ and the inclusion comes from the choice of $\epsilon$.

2. $\arg\min \ell(q', .) \neq \{z\}$, since the infimum in the definiton of $\alpha_0$ is a minimum.

Let $z' \in \arg\min \ell(q', .)$ with $z' \neq z$. Since $\arg\min \ell(q', .)$ is connected by (A4), Lemma 36 shows that there is a path of adjacent transpositions between $z$ and $z'$ in $\arg\min \ell(q', .)$. In particular, there exists two items $k', l'$ adjacent in $z$ such that $\tau_{k',l'} z \in \arg\min \ell(q', .)$ and $\tau_{k',l'} z \neq z$. The result follows from the three additional remarks:

1. We necessarily have $k' = k$ or $l' = k$.

   Indeed, aiming for a contradiction, assume $k \notin \{k', l'\}$. By the definition of $q_{\text{top}}^{(k)}$, we have $\ell(q_{\text{top}}^{(k)}, \tau_{k'l'} z) = \ell(q_{\text{top}}^{(k)}, z)$, and thus

$$\begin{aligned}
\ell(q', \tau_{k'l'} z) &= (1 - \alpha_0)\ell(q, \tau_{k'l'} z) + \alpha_0 \ell(q_{\text{top}}^{(k)}, z) \quad\quad\quad\quad\quad\quad (6) \\
&> (1 - \alpha_0)\ell(q, z) + \alpha_0 \ell(q_{\text{top}}^{(k)}, z) \quad\quad \text{because } \tau_{k'l'} z \notin \arg\min \ell(q, .) \\
&= \ell(q', z)
\end{aligned}$$

   which contradicts $\tau_{k'l'} z \in \arg\min \ell(q', .)$.

2. w.l.o.g., let $k' = k$. Then $\ell(q', \tau_{kl'} z) = \ell(q', \tau_{kl} z)$ and thus $\tau_{kl} z \in \arg\min \ell(q', .)$.

   Indeed, since $\tau_{k'l'} z \in \arg\min \ell(q', .)$, this means $\ell(q_{\text{top}}^{(k)}, \tau_{kl'} z) < \ell(q_{\text{top}}^{(k)}, z)$ and thus $z^{-1}(l') < z^{-1}(k)$ by the definition of $q_{\text{top}}^{(k)}$. Since $k$ and $l'$ are adjacent in $z$, and the rank of $l'$ is strictly smaller than that of $l$, we have $z^{-1}(l') = z^{-1}(l)$ and thus $l \sim_z l'$. This implies $\ell(q', \tau_{kl'} z) = \ell(q', \tau_{kl} z)$.

3. $\forall z'', \ell(q', z'') - \ell(q', \tau_{ij} z'') = (1 - \alpha_0)\big(\ell(q, z'') - \ell(q, \tau_{ij} z'')\big)$.

   This follows from the same calculation as (6), using the definition of $q_{\text{top}}^{(k)}$ and $k \notin \{i, j\}$. This remark implies that strict inequalities between $\ell(q, z'')$ and $\ell(q, \tau_{ij} z'')$ are preserved in $q'$.

The final result immediately follows, using the tie breaking lemma (Lemma 15) to find some other distribution for which $\tau_{k'l} z$ is the unique element of the argmin.

**Second case: $k \in \{i, j\}$.** Then, by assumption we have $l \neq i$ and $l \neq j$. In that case, we follow similar steps, using $q_{\text{bottom}}^{(l)}$ instead of $q_{\text{top}}^{(k)}$. $\qquad\qquad\qquad\qquad\qquad\qquad\qquad\qquad\quad\square$

## C.3 Strict monotonicity for losses with connected argmin

This section provides the final proof. The proof of the main result (coonected argmins are equivalent to the existence of a utility function) is based on the argument of *strict monotocity* of task losses with connected argmins (Lemma 40 and Corollary 41). In essence, the strict monotonicity extends to the entire task loss the property of its argmin, when these are connected: whenever item $i$ is preferred

to item $j$ in the argmin, it is always preferred. The strict monotocity allows to directly prove the existence of a utility function (Theorem 42). The utility function itself proves the calibration of the square loss, which in turn implies the connectedness of argmins using the characterization of calibration based on equality of argmins (Theorem 28). The strict monotonicity also readily implies connectedness of all sublevel sets, since it shows we can find a path between any permutation and an optimal permutation that never increases the task loss.

Before going to the proofs of strict monotonicity, we give the following alternative to the tie-breaking lemma (Lemma 15)[3]:

**Lemma 39.** *Under assumptions (A1), (A2) and (A3):*

*Let $q \in \mathcal{Q}$ and $i, j, z'$ such that $i \succ_{z'} j$. If $\ell(q, z') = \ell(q, \tau_{ij} z')$, then $\exists q' \in \mathcal{Q}$, such that $\operatorname{argmin} \ell(q', .) \subseteq \operatorname{argmin} \ell(q, .)$ and $\ell(q', z') > \ell(q', \tau_{ij} z')$.*

*Proof.* Let $\epsilon = \min_{z_1, z_2 : \ell(q, z_1) \neq \ell(q, z_2)} |\ell(q, z_1) - \ell(q, z_2)|$, take $\alpha$ such that $0 < \alpha < \frac{\epsilon}{\epsilon + \max_{z_0} \ell(q, z_0)}$, and define $q' = (1 - \alpha) q + \alpha q^{(\tau_{ij} z')}$ (as defined in Lemma 14). The choice of $\alpha$ makes sure that any strict preferences between any $z_1$ and $z_2$ by $\ell(q, .)$ are preserved in $\ell(q', .)$. Only ties in $\ell(q, .)$ can be changed in strict inequalities in $\ell(q', .)$. Thus $\operatorname{argmin} \ell(q', .) \subseteq \operatorname{argmin} \ell(q, .)$. Moreover, since $\ell(q^{(\tau_{ij} z')}, \tau_{ij} z') < \ell(q^{(\tau_{ij} z')}, , z')$ by definition of $q^{(\tau_{ij} z')}$, we have $\ell(q, z') > \ell(q, \tau_{ij} z')$, which is the desired result. □

We decribe the *strict monotonicity* property of losses with connected argmins (A4) in the next results.

**Lemma 40** (Strict monotonicity, base case). *Under assumptions (A1), (A2) and (A3) and (A4), we have:*

*$\forall q, z$ s.t. $\operatorname{argmin} \ell(q, .) = \{z\}$: $\forall i, j$ such that $i \succ_z j$, $\forall z'$ such that $i \succ_{z'} j$, $\ell(q, z') < \ell(q, \tau_{ij} z')$.*

*Proof.* Let $i, j$ such that $i \succ_z j$ and $z'$ such that $i \succ_{z'} j$. By Lemma 37, there is a path of adjacent transpositions between $z$ and $z'$ that never swaps $i$ and $j$. Let us denote by $z_0, ..., z_M$ such a path (with $z_0 = z$ and $z_M = z'$). Since we never swap $i$ and $j$, denoting $k_m, k_{m'}$ the adjacent items of $z_{m-1}$ that are swapped between $z_{m-1}$ and $z_m$, we have $\{i, j\} \neq \{k_m, k'_m\}$. We can thus apply Lemma 38, and find $q_0 = q, q_1, ..., q_M$ such that:

1. $\forall m \in [M], \quad \operatorname{argmin} \ell(q_m, .) = \{z_m\}$

2. $\forall m \in [M], \quad \forall z''$ such that $\ell(q_{m-1}, z'') > \ell(q_{m-1}, \tau_{ij} z'')$, we have $\ell(q_m, z'') > \ell(q_m, \tau_{ij} z'')$

An immediate consequence is that

$$\ell(q, z') \leq \ell(q, \tau_{ij} z').$$

Indeed, aiming for a contradiction, assume that $\ell(q, z') > \ell(q, \tau_{ij} z')$. Then by point 2. above, the sign of the difference would be kept along the path, all the way to $q_M$, i.e. we should have $\ell(q_M, z') > \ell(q_M, \tau_{ij} z')$. But $z' \in \operatorname{argmin} \ell(q_M, .)$, so this impossible.

Now, by Lemma 39, if we had $\ell(q, z') = \ell(q, \tau_{ij} z')$, we could find $q''$ with $\operatorname{argmin} \ell(q'', .) = \{z\}$ and $\ell(q'', z') > \ell(q'', \tau_{ij} z')$, which is impossible as we just stated. Thus, we have $\ell(q, z') < \ell(q, \tau_{ij} z')$. □

The lemma above extends to all possible $q$, not only those for which the argmin is a single element, in the following way:

**Corollary 41** (Strict monotonicity). *Under the assumptions of Lemma 40, let $q \in \mathcal{Q}$ and $i, j \in [n]$. There are three cases:*

i) *if $\forall z \in \operatorname{argmin} \ell(q,.), i \succ_z j$, then $\forall z' : i \succ_z j, \ell(q,z') < \ell(q,\tau_{i,j}z')$;*

ii) *if $\forall z \in \operatorname{argmin} \ell(q,.), i \succeq_z j$ and $\exists z_0 \in \operatorname{argmin} \ell(q,.), i \succ_{z_0} j$, then:*

$$\forall z' : i \succ_{z'} j, \ell(q,z') \leq \ell(q,\tau_{i,j}z').$$

*(Notice in that case $\ell(q,z_0) < \ell(q,\tau_{i,j}z_0)$.)*

iii) *if $\exists z_0, z_1 \in \operatorname{argmin} \ell(q,.)$ such that $i \succ_{z_0} j$ and $j \succ_{z_1} i$, then $\forall z' : \ell(q,z') = \ell(q,\tau_{i,j}z')$.*

*Proof.*

i) Let $q$ such that $\forall z \in \operatorname{argmin} \ell(q,.), i \succ_z j$. Aiming for a contradiction, assume $\ell(q,z') > \ell(q,\tau_{i,j}z')$. Let $z \in \operatorname{argmin} \ell(q,.)$. By the tie-breaking lemma (Lemma 15), we could find $q'$ with $\operatorname{argmin} \ell(q',.) = \{z\}$ and $\ell(q',z') > \ell(q',\tau_{i,j}z')$, but since $i \succ_z j$ this contradicts the base case (Lemma 40). We thus have $\ell(q,z') \leq \ell(q,\tau_{i,j}z')$. However, by Lemma 39, $\ell(q,z') = \ell(q,\tau_{i,j}z')$ would also yield a contradiction. We thus have $\ell(q,z') < \ell(q,\tau_{i,j}z')$, which is the desired result.

ii) $\forall z \in \operatorname{argmin} \ell(q,.), i \succeq_z j$ and $\exists z_0 \in \operatorname{argmin} \ell(q,.), i \succ_{z_0} j$. With the same arguments as above, taking $z \in \operatorname{argmin} \ell(q,.)$ such that $i \succ_z j$ (which exists by assumption) when using the tie-breaking lemma, we obtain $\forall z', \ell(q,z') \leq \ell(q,\tau_{i,j}z')$.

iii) Let $q$ such that $\exists z_0, z_1 \in \operatorname{argmin} \ell(q,.)$ such that $i \succ_{z_0} j$ and $j \succ_{z_1} i$. If there is $z'$ such that $i \succ_z j, \ell(q,z') > \ell(q,\tau_{i,j}z')$ then we can use the tie-breaking lemma with $z_0$ and that would contradicts the base case Lemma 40. Likewise, a $z'$ such that $i \succ_z j, \ell(q,z') < \ell(q,\tau_{i,j}z')$ would contradict the base case after applying the tie breaking lemma with $z_1$. Thus, equality must hold for all $z'$.

$\square$

The following result is a direct consequence of the strict monotonicity, and is the main result of the paper: there is a utility function wuch that the expected utility gives an optimal scoring function:

**Theorem 42.** *Let $\tilde{u} : \mathcal{Y} \to \mathbb{R}^n$ defined as: $\tilde{u}_i(y) = - \displaystyle\sum_{z \in \mathcal{Z} : z^{-1}(i)=1} L(y,z).$*

*For $q \in \mathcal{Q}$, denote the expected utility by $\tilde{U}(q) = \displaystyle\sum_{y \in \mathcal{Y}} q_y \tilde{u}(y).$*

*Under assumptions (A1), (A2) and (A3) and (A4), we have:*

$$\operatorname{argmin} \ell(q,.) = \operatorname{pred}(\tilde{U}(q))$$

Obviously, utilities are cardinal, in the sense that any affine transformation of $u$ is also a utility.

*Proof.* Let $q \in \mathcal{Q}$ and $i,j \in [n]$. From the definition of $u$ and the linearity (w.r.t. $q$) of $\tilde{U}$, we have:

$$\tilde{U}_i(q) - \tilde{U}_j(q) = \sum_{z \in \mathcal{Z} : z^{-1}(j)=1} \ell(q,z) - \sum_{z \in \mathcal{Z} : z^{-1}(j)=1} \ell(q,z)$$

$$= \sum_{z \in \mathcal{Z} : z^{-1}(i)=1} \left( \ell(q,\tau_{ij}z) - \ell(q,z) \right).$$

because $\{\tau_{ij}z \in \mathcal{Z} : z^{-1}(j)=1\} = \{z \in \mathcal{Z} : z^{-1}(i)=1\}$. Thus, coming back to the three cases of Lemma 41, we immediately have:

i) if $\forall z \in \operatorname{argmin} \ell(q,.), i \succ_z j$, then $\tilde{U}_i(q) > \tilde{U}_j(q)$;

ii) if $\forall z \in \operatorname{argmin} \ell(q,.), i \succeq_z j$ and $\exists z_0 \in \operatorname{argmin} \ell(q,.), i \succ_{z_0} j$, then $\tilde{U}_i(q) > \tilde{U}_j(q)$;

(The strict inequality comes from $\ell(q,z_0) < \ell(q,\tau_{i,j}z_0)$.)

iii) if $\exists z_0, z_1 \in \operatorname{argmin} \ell(q,.)$ such that $i \succ_{z_0} j$ and $j \succ_{z_1} i$, then $\tilde{U}_i(q) = \tilde{U}_j(q)$

The second and main step of the theorem is to show $\operatorname{argsort}(\tilde{U}(q)) \subseteq \bigcup_{z \in \operatorname{argmin} \ell(q,.)} z$. The main arguments consist in the following disjunction of cases. Denoting $\bigcup \operatorname{argmin} \ell(q,.) = \bigcup_{z \in \operatorname{argmin} \ell(q,.)} z$, we have:

    a) if $\forall z \in \operatorname{argmin} \ell(q,.), i \succ_z j$, then $\forall \sigma \in \bigcup \operatorname{argmin} \ell(q,.), i \succ_\sigma j$;

    b) if $\forall z \in \operatorname{argmin} \ell(q,.), i \succeq_z j$ and $\exists z_0 \in \operatorname{argmin} \ell(q,.), i \succ_{z_0} j$, then $\forall \sigma \in \bigcup \operatorname{argmin} \ell(q,.)$ we either have $i \succ_\sigma j$ or $\tau_{i,j}\sigma \in \bigcup \operatorname{argmin} \ell(q,.)$.

    c) if $\exists z_0, z_1 \in \operatorname{argmin} \ell(q,.)$ such that $i \succ_{z_0} j$ and $j \succ_{z_1} i$, then $\forall \sigma \in \bigcup \operatorname{argmin} \ell(q,.)$, we have $\tau_{ij}\sigma \in \bigcup \operatorname{argmin} \ell(q,.)$ (because $\ell(q, \Lambda(\sigma)) = \ell(q, \Lambda(\tau_{ij}\sigma))$ by *iii)* of Corollary 41).

    d) if $\forall z \in \operatorname{argmin} \ell(q,.), i \sim_z j$, then $\forall \sigma \bigcup \operatorname{argmin} \ell(q,.)$, we have $\tau_{ij}\sigma \in \bigcup \operatorname{argmin} \ell(q,.)$ (by definition of $\sim_z$).

The critical implication of points *a-d)* above is that for all $\sigma \in \bigcup \operatorname{argmin} \ell(q,.)$, if $\tilde{U}_i(q) > \tilde{U}_i(q)$ then $\sigma$ either gives a relative ordering of $i, j$ properly, or $\tau_{ij}\sigma \in \bigcup \operatorname{argmin} \ell(q,.)$. If $\tilde{U}_i(q) = \tilde{U}_i(q)$, which happens only in cases *c)* and *d)* above, then $\{\sigma, \tau_{ij}\sigma\} \subseteq \bigcup \operatorname{argmin} \ell(q,.)$. Successive applications of this remark allows us to construct $\sigma \in \operatorname{argsort}(\tilde{U}(q)) \cap \bigcup_{z \in \operatorname{argmin} \ell(q,.)}$: start from an arbitrary $\sigma \in \bigcup_{z \in \operatorname{argmin} \ell(q,.)}$, and take any $\sigma' \in \operatorname{argsort}(\tilde{U}(q))$. Apply bubble sort starting from $\sigma$ with target $\sigma'$. By the remark above, every step of bubble sort stays in $\bigcup \operatorname{argmin} \ell(q,.)$, which proves $\sigma' \in \bigcup \operatorname{argmin} \ell(q,.)$, and thus $\operatorname{argsort}(\tilde{U}(q)) \subset \bigcup \operatorname{argmin} \ell(q,.)$.

We thus proved $\forall q \in \mathcal{Q}, \operatorname{pred}(\tilde{U}q) \subset \operatorname{argmin} \ell(q,.)$. Equality is proved since this inclusion proves the square loss $y, s \mapsto (s - \tilde{u}(z))^2$ is $L$-calibrated, so equality follows from equality of argmins (Theorem 28).

$\square$

Now, we have all the tools to wrap up the proof of Theorem 32.

*Proof of Theorem 32.*
**(i) $\Rightarrow$ (ii).** As $L$ is CEU, there exists $u$ such that $\Phi_u^{\mathrm{sq}}$ is $L$-calibrated. Thanks to Th. 28, we know

$$\forall q \in \mathcal{Q}, \exists \delta_0 > 0, \forall 0 < \delta < \delta_0, \operatorname{pred}(\operatorname{lev}_\delta \phi_u^{\mathrm{sq}}(q,.)) = \operatorname{argmin} \ell(q,.)$$

As $\Phi_u^{\mathrm{sq}}$ is convex in $s$ and thus $\operatorname{lev}_\delta \phi_u^{\mathrm{sq}}(q,.)$ is connected, then $\forall q \in \mathcal{Q}, \operatorname{argmin} \ell(q,.)$ is connected.

**(ii) $\Rightarrow$ (i).** This is exactly Th. 42.

**(ii) $\Rightarrow$ (iii).** By Cor. 41, we have the property we referred to as *strict monotonicity* of the loss $L$. We use Lemma 36 to prove the connectedness by using path of adjacent items. Let us take $q \in \mathcal{Q}, \varepsilon > 0$ and $z \in \operatorname{lev}_\varepsilon \ell(q,.)$ and $\sigma \in z$. We know there exists $s \in \mathbb{R}^n$ such that $\operatorname{argmin} \ell(q,.) = \operatorname{pred}(s)$ (from (ii) $\Rightarrow$ (i)), thus we can find $\nu \in \mathfrak{S}_n$ such that $\operatorname{pred}(\nu) \in \operatorname{argmin} \ell(q,.)$ and for any $i, j$ such that $\tau_{ij} \in \texttt{bubblesort(target=}\nu\texttt{, input=}\sigma\texttt{)}$, we have $\forall z' \in \operatorname{argmin} \ell(q,.), i \prec_z j$. Indeed, if there were $\tau_{ij} \in \texttt{bubblesort(target=}\nu\texttt{, input=}\sigma\texttt{)}$ such that $\exists z' \operatorname{argmin} \ell(q,.), i \succeq_z j$ because $\operatorname{argmin} \ell(q,.) = \operatorname{pred}(s)$ then $\tau_{ij}\nu \in \operatorname{argsort}(s)$ and we could choose $\tau_{ij}\nu$ instead of $\nu$.

Denoting $(\tau^{(m)})_{m=1}^M$ the output of $\texttt{bubblesort(target=}\nu\texttt{, input=}\sigma\texttt{)}$, we just prove $\tau^{(1)}z \in \operatorname{lev}_\varepsilon \ell(q,.)$, and then finish by induction.
*Case 1: $i \sim_z j$.* In this case $\tau_{ij}z = z \in \operatorname{lev}_\varepsilon \ell(q,.)$.
*Case 2: $i \succ_z j$* By Cor. 41 *(i)*, $\ell(q, \tau_{ij}z) > \ell(q, z)$. Hence $\tau_{ij}z \in \operatorname{lev}_\varepsilon \ell(q,.)$. Then, by induction, for any $m \in [M]$ we have $\tau^{(m)} \ldots \tau^{(1)}z \in \operatorname{lev}_\varepsilon \ell(q,.)$.
Finally, any $z \in \operatorname{lev}_\varepsilon \ell(q,.)$ is connected to an element of $\operatorname{argmin} \ell(q,.)$ which is itself a connected set. Thus, $\operatorname{lev}_\varepsilon \ell(q,.)$ is connected.

**(iii) $\Rightarrow$ (ii).** Because $\forall q \in \mathcal{Q}, \exists \varepsilon > 0, \operatorname{lev}_\varepsilon \ell(q,.) = \operatorname{argmin} \ell(q,.)$. $\square$

# D   Proof of Corollary 7

*Disclaimer: As mentioned in the body of the paper, we address, in these Appendices, a setting more general than ranking. This setting is described in Section A. We restate first the statement of the paper (on ranking), then state the more general statement (on weak orders) and prove the latter.*

This section provides the proof of Corollary 7. First, we remind the exact statement of the paper in the ranking setting.

**Corollary 7.** *A ranking loss $L$ is CEU if and only if: for every distribution $P$ over $\mathcal{X} \times \mathcal{Y}$ such that $x \mapsto P(.|x)$ is continuous, there is a* continuous *optimal scoring function for $\mathcal{R}_{L,P}$.*

Then, the version of the same statement in $\mathcal{Z}$,

**Corollary 43.** *Under (A1), (A2) and (A3), the task loss $L$ is CEU if and only if for every distribution $P$ over $\mathcal{X} \times \mathcal{Y}$ such that $x \mapsto P(.|x)$ is continuous, there is a* continuous *optimal scoring function for $\mathcal{R}_{L,P}$ – i.e. a function $f : \mathcal{X} \to \mathbb{R}^n$ continuous such that $\forall x \in \mathcal{X}$, $\mathrm{pred}(f(x)) \subseteq \arg\min \ell(P(.|x), .)$.*

The proof of that corollary, as well as all proofs regarding local minima when there are disconnected argmins rely on the following path between probability distributions $\bar{q}$. The proof is straightforward.

**Lemma 44.** *Let $q_0$ such that $|\arg\min \ell(q_0, .)| > 1$. Let $z, z' \in \arg\min \ell(q_0, .)$ with $z \neq z'$. Define the following path between probability distributions in $\mathcal{Q}$:*

$$\forall \alpha \in [0,1], \quad \bar{q}(\alpha) = \begin{cases} (1 - 2\alpha)q^{(z)} + 2\alpha q_0 & \text{if } \alpha \in [0, \frac{1}{2}] \\ (2 - 2\alpha)q_0 + (2\alpha - 1)q^{(z')} & \text{if } \alpha \in [\frac{1}{2}, 1] \end{cases}.$$

*Then, let*

$$\epsilon_0 = \min_{q' \in \{q_0, q^{(z)}, q^{(z')}\}} \min_{z'' \notin \arg\min \ell(q', .)} \left( \ell(q', z'') - \underline{\ell}(q') \right)$$

*Then, $\epsilon > 0$ and we have:*

$$\forall \alpha \in [0,1], \forall z'' \in \mathrm{lev}_{\epsilon_0} \ell(\bar{q}(\alpha), .), z'' \in \arg\min \ell(q_0, .).$$

*Proof.* We prove the case $\alpha \in [0, \frac{1}{2}]$, the other case is similar. Let $\alpha \in [0, \frac{1}{2}]$. We first notice $z \in \arg\min \ell(\bar{q}(\alpha), .)$. Thus, by developping $\bar{q}(\alpha)$ we have, for $z'' \neq z$:

$$\ell(\bar{q}(\alpha), z'') - \underline{\ell}(\bar{q}(\alpha)) = (1 - 2\alpha)\left( \ell(q^{(z)}, z'') - \ell(q^{(z)}, z) \right) + 2\alpha\left( \ell(q_0, z'') - \ell(q_0, z) \right) \geq \epsilon. \qquad \square$$

*Proof of Corollary 43.* The direct implication is straightforward: if $L$ is CEU, we can choose $f : x \mapsto \mathbb{E}_{P(.|x)}[u(Y)]$ as optimal scoring function by Theorem 32, which is continuous whenever $x \mapsto P(.|x)$ is continuous.

For the reverse implication, let $q_0 \in \mathcal{Q}$. We have to prove that $\arg\min \ell(q_0, .)$ is connected. Notice that if $|\arg\min \ell(q_0, .)| = 1$ it is connected, so we focus on the case $|\arg\min \ell(q_0, .)| > 1$. Let $z, z' \in \arg\min \ell(q_0, .)$ with $z \neq z'$. Let $\mathcal{X} = [0, 1]$, and take the distribution $P$ over $\mathcal{X} \times \mathcal{Y}$ such that the marginal distribution over $\mathcal{X}$ is uniform and $x \mapsto P(.|x)$ is the path $\bar{q}$ constructed as in Lemma 44. If there is a continuous optimal scoring function $f : [0, 1] \to \mathbb{R}^n$, it means that $\forall \alpha \in [0, 1]$, $\mathrm{pred}(f(\alpha)) \in \arg\min \ell(\bar{q}(\alpha), .)$ and thus $\mathrm{pred}(f([0, 1])) \subseteq \arg\min \ell(q_0, .)$ by Lemma 44.

Notice that $f$ is continuous and thus preserves connectedness, so $\mathrm{pred}(f([0, 1]))$ is connceted in $\mathcal{Z}$ by definition. By the characterization of connectedness in $\mathcal{Z}$ through paths of adjacent transpositions (Theorem 36), there is a path between $z$ and $z'$ in $\arg\min \ell(q_0, .)$. The construction above can be repeated for every $z, z' \in \arg\min \ell(q_0, .)$, which implies that $\arg\min \ell(q_0, .)$ is connected. $\qquad \square$

# E   Local Minima of the Task Loss

## E.1   Proof of Theorem 46

**Definition 8.** *Given a distribution $q \in \mathcal{Q}$ and a loss $L$, a ranking $\sigma \in \mathfrak{S}_n$ is a* local minimum *if for any $r \in [n-1]$, $\ell(q, \sigma) \leq \ell(q, \sigma\tau_{r,r+1})$.*

The definition of local minima for ranking losses 8 is extended to $\mathcal{Z}$ in a straightforward manner:

**Definition 45.** *Given a loss $L$ and $q \in \mathcal{Q}$, a prediction $z \in \mathcal{Z}$ is a local minimum if, for any pair of adjacent items $(i, j)$, we have $\ell(q, z) \leq \ell(q, \tau_{ij} z)$.*

**Theorem 46.** *Under (A1), (A2) and (A3), if a task loss $L$ is not CEU, then the subset of distribution for which $\ell$ has non-global local minima has non-zero measure.*

*Proof.* We use the following property of the excess inner risk, where we denote $\|L\|_\infty = \max_{y,z} |L(y, z)|$. The calculation is similar to the one later in Lemma 48 for surrogate losses.

$$\forall q, q', \forall z, \ \ \left| \ell(q', z) - \underline{\ell}(q') - \left( \ell(q, z) - \underline{\ell}(q) \right) \right| \leq 2\|q - q'\|_1 \|L\|_\infty.$$

Let us take $q_0$ such that $\operatorname{argmin} \ell(q_0, .)$ is not connected, and $z \in \operatorname{argmin} \ell(q_0, .)$.

Let $\bar{q}(\alpha) = (1 - \alpha)q_0 + \alpha q^{(z)}$. We define the *gap* with respect to $q_0$ of $q \in \mathcal{Q}$ as:

$$G(q) = \min_{z' \notin \operatorname{argmin} \ell(q_0, .)} \left( \ell(q, z') - \underline{\ell}(q) \right).$$

We use $\epsilon_0 = G(q_0)$.

The important aspect of this gap is that if $z$ and $z'$ are both in $\operatorname{argmin} \ell(q_0, .)$ but not connected in $\operatorname{argmin} \ell(q_0, .)$, this disconnectedness remains in sublevel sets of $\ell(q, .)$ for other distributions $q$: If, for some $\epsilon > 0$, $z$ and $z'$ are not connected in $\operatorname{lev}_\epsilon \ell(q, .)$ with $\epsilon \leq G(q)$, then $z$ and $z'$ are in different connected components of $\operatorname{lev}_\epsilon \ell(q, .)$. We prove the existence of bad local minima by showing that there are suboptimal connected components smaller than the gap.

Let $\alpha_0 \in (0, 1]$. We then have $\{z\} = \operatorname{argmin} \ell(\bar{q}(\alpha_0), .)$. Moreover, let $\tilde{\epsilon} = \min_{z' \neq z} (\ell(q^{(z)}, z') - \underline{\ell}(q^{(z)}))$. Notice that $\tilde{\epsilon} > 0$ by definition of $q^{(z)}$. We have[4]:

$$G(\bar{q}(\alpha_0)) \geq \epsilon_0 - 4\alpha_0 \|L\|_\infty,$$
$$\forall z' \in \operatorname{argmin} \ell(q_0, .), z' \neq z, \quad 4\alpha_0 \|L\|_\infty \geq \ell(\bar{q}(\alpha_0), z') - \underline{\ell}(\bar{q}(\alpha_0)) \geq \alpha_0 \tilde{\epsilon}.$$

And thus, given $\eta > 0$, for every $q'$ such that $\|q' - \bar{q}(\alpha_0)\|_1 \leq \eta$, we have

$$G(q') \geq \epsilon_0 - (4\alpha_0 + 2\eta)\|L\|_\infty,$$

and $\forall z' \in \operatorname{argmin} \ell(q_0, .), z' \neq z$:

$$(4\alpha_0 + 2\eta)\|L\|_\infty \geq \ell(q', z') - \underline{\ell}(q') \geq \alpha_0 \tilde{\epsilon} - 2\eta\|L\|_\infty.$$

Let $\alpha_0 \in (0, 1)$ and $\eta \in (0, 1)$ such that

$$0 < \alpha_0 \tilde{\epsilon} - 2\eta\|L\|_\infty < (4\alpha_0 + 2\eta)\|L\|_\infty < \epsilon_0 - (4\alpha_0 + 2\eta)\|L\|_\infty.$$

These exist since $\epsilon_0 > 0$. Then, for any $q'$ such that $\|\bar{q}(\alpha_0) - q'\| < \eta$, for any $z' \in \operatorname{argmin} \ell(q_0, .)$ such that $z'$ is not in the same connected component as $z$, with such values of $\alpha_0$ and $\eta$, $z'$ is in a connected component of $\operatorname{lev}_{\epsilon_0 - (4\alpha_0 + 2\eta)\|L\|_\infty} \ell(q', .)$ that is suboptimal (because $\alpha_0 \tilde{\epsilon} - 2\eta\|L\|_\infty > 0$) and disconnected from $z$ (becuase $G(q') \geq \epsilon_0 - (4\alpha_0 + 2\eta)\|L\|_\infty$). Thus, all such $q'$ have a bad local minimum. The result follows, since the measure of the $\|.\|_1$-ball of radius $\eta$ is non-zero. $\square$

# F   Proof of Utility Computation on Generalized DCG

The expression of the utility $u$ from Theorem 6 may not be efficient to compute a priori. As it happens, for many evaluation metrics, the expression of $u$ does simplify for numerous common tasks losses.

$$\ell(\bar{q}(\alpha_0), z') - \underline{\ell}(\bar{q}(\alpha_0)) = \alpha_0 \left( \ell(q^{(z)}, z') - \ell(q^{(z)}, z) \right) + (1 - \alpha_0)\left( \ell(q_0, z') - \ell(q_0, z) \right).$$

where the term in $q_0$ vanishes because both $z$ and $z'$ are optimal for $q_0$.

**Proposition 47.** *We assume here a ranking task loss $L$ of the form $L(y, \sigma) = \mathrm{DCG}_{w,u}(y, \sigma) = -\sum_{k=1}^{n} w_k u_{\sigma(k)}(y) - b(y)$ where $\forall i \in [n], u_i$ is increasing and $w$ is decreasing. Then, there exists $B : \mathcal{Y} \to \mathbb{R}_*^+$ and $A > 0$ such that, $\forall i \in [n], \forall y \in \mathcal{Y}$,*

$$\tilde{u}_i(y) = B(y) + A u_i(y)$$

*Proof.* By definition, we have,

$$u_i(y) = -\sum_{\sigma \in \mathfrak{S}_n} \mathbb{1}_{[\sigma(1)=i]} L(y, \sigma)$$

$$= \sum_{\sigma \in \mathfrak{S}_n} \mathbb{1}_{[\sigma(1)=i]} \left( b(y) + \sum_{r=1}^{n} u_{\sigma(r)}(y) w_r \right)$$

$$= b(y)(n-1)! + \sum_{\sigma \in \mathfrak{S}_n} \sum_{r=1}^{n} \mathbb{1}_{[\sigma(1)=i]} u_{\sigma(r)}(y) w_r$$

$$= b(y)(n-1)! + \textcolor{red}{u_i(y) w_1 (n-1)!} + \sum_{\sigma \in \mathfrak{S}_n} \sum_{r=2}^{n} \mathbb{1}_{[\sigma(1)=i]} u_{\sigma(r)}(y) w_r$$

$$= b(y)(n-1)! + u_i(y) w_1 (n-1)! + \sum_{\sigma \in \mathfrak{S}_n} \sum_{r=2}^{n} \textcolor{red}{\sum_{k \neq i}} \mathbb{1}_{[\sigma(r)=k]} \mathbb{1}_{[\sigma(1)=i]} u_k(y) w_r$$

$$= b(y)(n-1)! + u_i(y) w_1 (n-1)! + \sum_{k=1}^{n} \sum_{r=2}^{n} u_k(y) w_r \sum_{\sigma \in \mathfrak{S}_n} \mathbb{1}_{[\sigma(r)=k]} \mathbb{1}_{[\sigma(1)=i]}$$

$$= b(y)(n-1)! + u_i(y) w_1 (n-1)! + \sum_{k=1}^{n} \sum_{r=2}^{n} u_k(y) w_r \mathbb{1}_{[k \neq i]} (n-2)!$$

$$= b(y)(n-1)! + u_i(y) w_1 (n-1)! \textcolor{red}{- \sum_{k=1}^{n} \sum_{r=2}^{n} u_k(y) w_r \mathbb{1}_{[k=i]} (n-2)!} + \sum_{k=1}^{n} \sum_{r=2}^{n} u_k(y) w_r \left( \mathbb{1}_{[k \neq i]} \textcolor{red}{+ \mathbb{1}_{[k=i]}} \right)(n-2)!$$

$$= b(y)(n-1)! + \sum_{r=2}^{n} \left( u_i(y) w_1 - u_i(y) w_r \right)(n-2)! + \sum_{k=1}^{n} \sum_{r=2}^{n} u_k(y) w_r (n-2)!$$

$$= \underbrace{b(y)(n-1)! + \sum_{k=1}^{n} \sum_{r=2}^{n} u_k(y) w_r (n-2)!}_{B(y)} + u_i(y) \underbrace{\sum_{r=2}^{n} \left( w_1 - w_r \right)(n-2)!}_{A}$$

$$= B(y) + A u_i(y)$$

$\square$

# G    Gumbel Smoothing is a Calibrated Surrogate Loss

*Disclaimer: While this section presents a way to build, for any $L$, a non-convex surrogate loss $L$-calibrated, it is by no means one to use in practice. Its interest is mostly theoretical as its computational complexity is generally prohibitive.*

*Disclaimer: For this section, we keep the specific case of the ranking where $\mathcal{Z} = \mathfrak{S}_n$.*

We examine here a convolution with a well-behaved kernel to smooth $L(., \mathrm{argsort}(.))$. We obtain the following surrogate loss,

$$\Phi_L^{\mathrm{NC}} : y, s \mapsto \int_{\mathbb{R}^n} L(y, \mathrm{argsort}(u - s)) \kappa(u) \mathrm{d}u$$

The particular kernel $\kappa$ we choose here is a Gumbel density as it can easily be reformulated in the ranking space using a Plackett-Luce model. Indeed, for any $s \in \mathbb{R}^n$ and $u \sim \mathrm{Gumbel}(0, 1)^n$,

we have that $\mathrm{argsort}(s + u)$ follows a Plackett-Luce distribution[38], allowing to express $\Phi_L^{\mathrm{NC}}$ in closed-form (with prohibitive computational cost for large values of $n$):

$$
\begin{aligned}
\Phi_L^{\mathrm{NC}}(y, s) &= \int_{\mathbb{R}^n} L(y, \mathrm{argsort}(s - u))\kappa(u)\mathrm{d}u \\
&= \int_{\mathbb{R}^n} L(y, \mathrm{argsort}(s - u))k(u)\mathrm{d}u \\
&= \sum_{\sigma \in \mathfrak{S}_n} L(y, \sigma) \underbrace{\int_{\mathbb{R}^n} \mathbb{1}_{[\mathrm{argsort}(s-u)=\sigma]}k(u)\mathrm{d}u}_{\mathbb{P}(\sigma|e^s) \text{ of a Plackett-Luce model}} \\
&= \sum_{\sigma \in \mathfrak{S}_n} L(y, \sigma) \prod_{r=1}^{n} \frac{e^{s_{\sigma(r)}}}{\sum_{k \geq r} e^{s_{\sigma(k)}}}
\end{aligned}
$$

**Proposition 11.** *For any ranking loss L, the following surrogate loss $\Phi_L^{\mathrm{NC}}$ is L-calibrated.*

$$
\Phi_L^{\mathrm{NC}} : y, s \mapsto \int_{\mathbb{R}^n} L(y, \mathrm{argsort}(u - s))\kappa(u)\mathrm{d}u = \sum_{\sigma \in \mathfrak{S}_n} L(y, \sigma) \prod_{r=1}^{n} \frac{e^{s_{\sigma(r)}}}{\sum_{k \geq r} e^{s_{\sigma(k)}}}
$$

*Proof.* Considering the surrogate $\Phi = \Phi_L^{\mathrm{NC}}$, let $q \in \mathcal{P}$, to show the calibration, we show that

1. $\underline{\phi}(q) = \underline{\ell}(q)$.

2. $\forall s \in \mathbb{R}^n$ such that $\exists \nu \in \mathrm{argsort}(s), \ell(q, \nu) > \underline{\ell}(q)$ we have $\phi(q, s) \geq \frac{n!-1}{n!}\underline{\ell}(q) + \frac{1}{n!}\ell(q, \nu) > \underline{\ell}(q)$

Let's prove 1. first. We can choose $\sigma \in \mathrm{argmin}\, \ell(q, .)$ and $s \in \mathbb{R}^n$ such that $\mathrm{argsort}(s) = \{\sigma\}$. Taking $\alpha > 0$, we have $\mathbb{P}(\sigma|e^{\alpha s}) \xrightarrow[\alpha \to \infty]{} 1$, meaning that $\phi(q, \alpha s) \xrightarrow[\alpha \to \infty]{} \ell(q, \sigma) = \underline{\ell}(q)$.

Let's prove 2. now and consider $s \in \mathbb{R}^n, \exists \nu \in \mathrm{argsort}(s), \ell(q, \nu) > \underline{\ell}(q)$. Because $\nu \in \mathrm{argsort}(s)$ it is an event of maximal probability in the Plackett-Luce model $\mathbb{P}(.|e^s)$. Hence $\mathbb{P}(\nu|e^s) > \frac{1}{n!}$.

$$
\begin{aligned}
\phi(q, s) &= \sum_{\sigma \in \mathfrak{S}_n} \ell(q, \sigma)\mathbb{P}(\sigma|e^s) \\
&\geq \frac{1}{n!}\ell(q, \nu) + \left(\mathbb{P}(\nu|e^s) - \frac{1}{n!}\right)\ell(q, \nu) + \sum_{\sigma \neq \nu} \ell(q, \sigma)\mathbb{P}(\sigma|e^s) \\
&\geq \frac{1}{n!}\ell(q, \nu) + \left(\mathbb{P}(\nu|e^s) - \frac{1}{n!}\right)\underline{\ell}(q) + \sum_{\sigma \neq \nu} \underline{\ell}(q)\mathbb{P}(\sigma|e^s) \\
&\geq \frac{1}{n!}\ell(q, \nu) + \frac{n!-1}{n!}\underline{\ell}(q)
\end{aligned}
$$

$\square$

# H   Local Minima of Surrogate Loss

## H.1   Simulations to Analyze Surrogate Bad Local Valleys

We describe here the simulations ran to analyze the loss surface of the $\Phi_L^{\mathrm{NC}}$. We remind its definition:

$$
\Phi_L^{\mathrm{NC}} : y, s \mapsto \int_{\mathbb{R}^n} L(y, \mathrm{argsort}(u - s))\kappa(u)\mathrm{d}u = \sum_{\sigma \in \mathfrak{S}_n} L(y, \sigma) \prod_{r=1}^{n} \frac{e^{s_{\sigma(r)}}}{\sum_{k \geq r} e^{s_{\sigma(k)}}}.
$$

Let us first make a few remarks on $\Phi_L^{\mathrm{NC}}$ to explain the intention and the design of the experiments. First, about the smoothing. It is common, when smoothing a function with a convolution, to have the strength of the smoothing controlled by the bandwidth of the convolution kernel (e.g., variance of a Gaussian kernel). This is unnecessary here: the function $s \mapsto L(y, \mathrm{argsort}(s))$ is invariant by

re-scaling of the scores $s$ (contrarily to the convolution kernel). Thus, the loss $L$ is more smoothed towards vectors $s$ of small norm (up to removing a translation) and almost not smoothed when the norm of $s$ goes towards infinity.

Second, because the task loss is more smoothed towards scores of low norm, initializing optimization algorithms close to 0 (all items have the same score) proves to be an empirically good heuristic.

We performed two experiments, similarly to the simulation for the task losses described in Section 3. Given $n$ items, we ran simulations where each simulation consists in sampling a distribution $q$ then, for each $q$, we randomly choose a starting point $s_0$ for the optimization and run a gradient descent on $\Phi_L^{\text{NC}}$ with a basic line search to handle the poor conditioning of $\Phi_L^{\text{NC}}$. To handle bad valleys where the infimum is not reached while avoiding numerical issues, we project the optimization to stay inside a $\|.\|_2$ ball of radius 100. This is sufficient to ensure that the local minimum reached in the $\|.\|_2$ ball generates the same ranking $\sigma$ as the scores at infinity. We then compute the sub-optimality of this local valley as $\eta = \frac{\ell(q,\sigma) - \min \ell(q,.)}{\max \ell(q,.) - \min \ell(q,.)}$.

**Simulation 3: Distribution of sub-optimality of bad local valleys.** Here, $q$ is sampled uniformly on $\mathcal{Q}$ and $s_0$ is sampled from a $\mathcal{N}(0, \text{Id}_n)$, where $\text{Id}_n$ is the $n \times x$ identity matrix. Figure 3 *(left and middle)* (reproduced in Figure 4) illustrates the results of the distribution of $\eta$ given that $\eta > 0$. In these plots, the suboptimality of *bad* local valleys are considered in the distribution.

Figure 4: Distribution of sub-optimality of bad local valleys. Left: ERR. Right:AP.

**Simulation 4: Percentage of optimization runs stuck in bad local valleys.** Here, we want to answer the question: *How often can we expect the optimization of the surrogate to be stuck in a bad local valley when the task loss has several local minima?* To answer this, $q$ is sampled uniformly amongst distributions for which $\ell(q,.)$ has at least two local minima, and $s_0$ is set to 0 as it proved an empirically good intialization. Figure 3 (right) (reproduced here as Figure 5) illustrates the percentage of runs that are stuck in local minima as a function of the number of items $n$ for the ERR and the AP. For $n = 3$ 5-10% of the runs are stuck in local minima, and is growing with the number of items $n$ until $n = 6$. Moreover, the bad local valleys of $\Phi_L^{\text{NC}}$ found, actually corresponds to rankings that are bad local minima for the task loss itself. This suggests that these local minima of the surrogate loss reflect an intrinsic difficulty of the optimization for calibrated surrogate losses rather than an artifact of $\Phi_L^{\text{NC}}$. We thus conjecture that optimization is difficult for other surrogate losses as well.

## H.2 Preliminaries for the study of surrogate losses

The two subsequent sections study properties of surrogate losses calibrated with non-CEU losses. We summarize here basic results that the two sections use.

We first summarize the assumptions we use:

>**(A5).** *The four following statements hold:*

> i) *(A1), (A2) and (A3) hold,*

Figure 5: Percentage of optimizations runs on $\Phi_L^{\mathrm{NC}}$ that end up stuck in a bad local valley over distribution for which the task loss as several local minima.

  ii) $\exists q_0$ such that $\operatorname{argmin} \ell(q_0, .)$ is not connected,

  iii) $\Phi : \mathcal{Y} \times \mathbb{R}^n \to \mathbb{R}$ is such that $\forall y \in \mathcal{Y}$, $s \mapsto \Phi(y, s)$ is $\beta_\Phi$-Lipschitz and uniformly bounded by $B_\Phi$,

  iv) $\Phi$ is L-calibrated.

The assumption above implies Lipschitzness of the excess inner risk $\phi(q, s) - \underline{\phi}(q)$ with respect to both $q \in \mathcal{Q}$ and $s \in \mathbb{R}^n$:

**Lemma 48.** *If $\Phi$ is bounded and Lipschitz as defined in Assumption (A5) iii), then:*

$$\forall q, q', \forall s, s', \quad |\phi(q, s) - \underline{\phi}(q) - \phi(q', s') - \underline{\phi}(q')| \leq 2\|q - q'\|_1 B_\Phi + \beta_\Phi \|s - s'\|_2$$

*Proof.* First consider $A = \phi(q, s) - \underline{\phi}(q) - \phi(q', s') - \underline{\phi}(q')$. Let $\epsilon > 0$ and $s^*$ such that $\underline{\phi}(q) \geq \phi(q, s^*) - \epsilon$. We have

$$A \leq \phi(q, s) - \phi(q, s^*) + \epsilon - \phi(q', s) + \phi(q', s^*)$$
$$\leq \|q - q'\|_1 \max_y |\Phi(y, s) - \Phi(y, s^*)| + \phi(q', s) - \phi(q', s) + \epsilon$$
$$\leq 2\|q - q'\|_1 B_\Phi + \beta_\Phi \|s - s'\|_2 + \epsilon,$$

where the first inequality holds since $\phi(q', s^*) \geq \underline{\phi}(q')$ and the second from Hölder's inequality. Following the same steps, we find the same upper bound for $\phi(q', s') - \underline{\phi}(q') - \phi(q, s) - \underline{\phi}(q)$. The final result is obtained in the limit $\epsilon \to 0$. $\square$

The following result give the main implications of calibration of $\Phi$. It are extended version of Lemma 44 for surrogate losses instead of task losses. It gives basic properties of segments joining a distribution $q_0$ with a disconnected argmin, with distributions of type $q^{(z)}$ where $z \in \operatorname{argmin} \ell(q_0, ., .)$. The lemma has several parts that are used in different proofs.

**Lemma 49.** *Under Assumption (A5), let $q_0 \in \mathcal{Q}$ such that $\operatorname{argmin} \ell(q_0, .)$ is disconnected.*

  i) *Let $z, z' \in \operatorname{argmin} \ell(q_0, .)$ such that $z$ and $z'$ are not in the same connected components of $\operatorname{argmin} \ell(q_0, .)$. Let $\bar{q}$ be defined as in Lemma 44. Then $\exists \delta_0$ such that for every continuous function $f : [0, 1] \to \mathbb{R}^n$ such that $\{z, z'\} \subseteq \operatorname{pred}(f([0, 1]))$, we have:*

$$\max_{\alpha \in [0,1]} \left( \phi(\bar{q}(\alpha), f(\alpha)) - \underline{\phi}(\bar{q}(\alpha)) \right) \geq \delta.$$

  ii) *Let $\bar{q} : [0, 1] \to \mathcal{Q}$ defined by:*

$$\forall \alpha \in [0, 1], \quad \bar{q}(\alpha) = (1 - \alpha)q_0 + \alpha q^{(z)}.$$

*Let $G_0 = \min\limits_{z \notin \mathrm{argmin}_{q_0}} \min\limits_{q \in \{q_0, q^{(z)}\}} \big(\ell(q, z) - \underline{\ell}(q)\big)$. Then $G_0 > 0$.*

*Moreover, let $\eta_0 \in (0, \frac{G_0}{2\|L\|_\infty})$. Then $\exists \delta_0 > 0$ such that:*

$$\forall \alpha \in [0, 1], \forall q \in \mathcal{B}_1(\bar{q}(\alpha), \eta), \quad \forall s \in \mathrm{lev}_{\delta_0}\phi(q, .), \mathrm{pred}(s) \subseteq \mathrm{argmin}\,\ell(q_0, .).$$

*iii) For any $q \in \mathcal{Q}$ such that $\mathrm{argmin}\,\ell(q, .) = \{z\}$, there is $\delta_0 > 0$ and $\eta_0 > 0$ such that:*

$$\forall s \in \mathrm{lev}_{\delta_0}\phi(q, .), \forall q' \in \mathcal{B}_1(q, \eta_0), \quad \mathrm{pred}(s) = \{z\}.$$

*Proof.* Point *i).* Using uniform calibration (Theorem 30, let $\epsilon_0$ be defined as in Lemma 44, and let $\delta_0$ such that $\forall q \in \mathcal{Q}, s \in \mathrm{lev}_{\delta_0}\ell(q, .), \mathrm{pred}(s) \subseteq \mathrm{lev}_{\epsilon_0}\ell(q, .)$. Combining with Lemma 44, we have:

$$\forall \alpha \in [0, 1], \forall s \in \mathrm{lev}_{\delta_0}\phi(\bar{q}(\alpha), .), \mathrm{pred}(s) \subseteq \mathrm{argmin}\,\ell(q_0, .). \tag{7}$$

Since connectedness is preserved by continuous functions, any continuous function $f : [0, 1] \to \mathbb{R}^n$ such that $\{z, z'\} \subseteq \mathrm{pred}(f([0, 1]))$ must have $\mathrm{pred}(f([0, 1])) \not\subseteq \mathrm{argmin}\,\ell(q_0, .)$. Using (7), this means $\exists \alpha, f(\alpha) \notin \mathrm{lev}_{\delta_0}\phi(\bar{q}(\alpha), .)$, which is equivalent to the desired result.

Point *ii).* With the same arguments as Lemma 44, for every $\alpha \in [0, 1]$, if $z \in \mathrm{lev}_{G_0}\ell(\bar{q}(\alpha), .)$ then $z \in \mathrm{argmin}\,\ell(q_0, .)$.

Now, let us take $\eta \in (0, \frac{G_0}{2\|L\|_\infty})$. For every $\alpha \in [0, 1]$, the Lipschitzness of $q \mapsto \ell(q, z) - \underline{\ell}(q)$, which follows from similar arguments as Lemma 48, gives:

$$\forall \alpha \in [0, 1], \forall q \in \mathcal{B}_1(\bar{q}(\alpha), \eta), \forall z \notin \mathrm{argmin}\,\ell(q_0, .), \; \ell(q, z) - \underline{\ell}(q) \geq G_0 - 2\eta\|L\|_\infty > 0 \tag{8}$$

By uniform calibration (Theorem 30), there is $\delta_0 > 0$ such that $\forall \delta \in (0, \delta_0), \forall q \in \mathcal{Q}, \forall s \in \mathrm{lev}_{\delta}\phi(q, .), \mathrm{pred}(s) \subseteq \mathrm{lev}_{G_0 - 2\eta\|L\|_\infty}\ell(q, .)$. With this choice of $\delta_0$, using (8) gives the result.

Point *iii).* The proof is exactly the same a before: Let $\epsilon = \min_{z' \neq z}(\ell(q, z') - \underline{\ell}(z))$. We have $\epsilon > 0$ by the assumption $\mathrm{argmin}\,\ell(q, .) = \{z\}$. Let $\eta_0 \in (0, \frac{\epsilon}{2\|L\|_\infty})$. We have

$$\forall q' \in \mathcal{B}_1(q, \eta), \min_{z' \neq z}(\ell(q', z') - \underline{\ell}(z)) \geq \epsilon - 2\eta\|L\|_\infty > 0.$$

And thus $\forall q' \in \mathcal{B}_1(q, \eta), \{z\} = \mathrm{argmin}\,\ell(q, .)'$. By uniform calibration (Th. 30), using $\delta_0 > 0$ such that $\forall q' \in \mathcal{Q}, \forall s \in \mathrm{lev}_{\delta_0}\phi(q', .), s \in \mathrm{lev}_{\epsilon - 2\eta\|L\|_\infty}\ell(q', .)$ gives the result. $\square$

## H.3 Lower bound on approximation error of surrogate loss by Lipschitz functions

The proposition below is a stronger version of Proposition 13. It makes it precise that the conditional distribution over $\mathcal{Y}$ is infinitely many times differentiable with bounded derivatives at any order (denoted $x \mapsto P(.|x) \in \mathcal{W}^\infty([0, 1]^\mathcal{Q}))$, and that there is a single point in $[0, 1]$ on which the argmin of $\ell$ is not a singleton (and thus disconnected) (condition $|\{\alpha \in [0, 1] : |\mathrm{argmin}\,\ell(p_{Y|X=\alpha}|, .)| > 1\}| = 1$). This makes sure that the distibution is not too peculiar.

**Proposition 50.** *Under Assumption (A5), there is a probability measure $P$ over $[0, 1] \times \mathcal{Y}$ where the marginal distribution over $[0, 1]$, and $x \mapsto P(.|x) \in \mathcal{W}^\infty([0, 1]^\mathcal{Q})$ such that $|\{\alpha \in [0, 1] : |\mathrm{argmin}\,\ell(p_{Y|X=\alpha}|, .)| > 1\}| = 1$, and constants $c, c' > 0$, such that for all $\beta \geq 0$*

$$\inf_{\substack{f:[0,1] \to \mathbb{R}^n \\ f:\beta - Lipschitz}} \mathcal{R}_{\Phi, P}(f) - \inf_{g:\mathcal{X} \to \mathbb{R}^n} \mathcal{R}_{\Phi, P}(g) \geq \min\big(c', \frac{c}{8B_\Phi + \beta_\Phi \beta}\big).$$

*Proof.* Let $q_0$ such that $\mathrm{argmin}\,\ell(q_0, .)$ is disconnected and let $z, z' \in \mathrm{argmin}\,\ell(q_0, .)$ such that $z$ and $z'$ belong to two different connected components of $\mathrm{argmin}\,\ell(q_0, .)$. Let $\bar{q}(\alpha)$ defined as in Lemma 44. Clearly, $\bar{q} \in \mathcal{W}^\infty([0, 1]^{|\mathcal{Y}|})$, i.e., $\bar{q}$ has bounded derivatives of any order. Also there is only one value of $\alpha$ ($\alpha = \frac{1}{2}$) for which $\mathrm{argmin}\,\ell(\bar{q}(\alpha), .)$ is not a singleton.

Let $f$ be $\beta$-Lipschitz. We consider two cases:

- If $\{z, z'\} \subset \mathrm{pred}(f([0,1]))$:

  First, we show that $\alpha \mapsto \phi(\bar{q}(\alpha), f(\alpha)) - \underline{\phi}(\bar{q}(\alpha))$ is Lipschitz. Notice that $\|\bar{q}(\alpha) - \bar{q}(\alpha')\|_1 \leq 2|\alpha - \alpha'|$ when $\alpha$ and $\alpha'$ are in the same help-segment of $[0,1]$. When they are not, say $\alpha \in [0, \frac{1}{2})$ and $\alpha' \in [\frac{1}{2}, 1]$, we have

  $$\|\bar{q}(\alpha) - \bar{q}(\alpha')\|_1 \leq \|(2 - \alpha - \alpha')q_0\|_1 + 2\left\|(\frac{1}{2} - \alpha)q^{(z)}\right\|_1 + 2\left\|(\frac{1}{2} - \alpha')q^{(z')}\right\|_1$$

  $$\leq 4(|\frac{1}{2} - \alpha| + |\frac{1}{2} - \alpha'|) = 4(\alpha' - \alpha).$$

  We thus have that $\alpha \mapsto \phi(\bar{q}(\alpha), f(\alpha)) - \underline{\phi}(\bar{q}(\alpha))$ is $8B_\Phi + \beta_\Phi \beta$-Lipschitz by Lemma 48.

  Moreover, by point *i)* of Lemma 49, choose $\delta > 0$ such that $\max_{\alpha \in [0,1]} \phi(\bar{q}(\alpha), f(\alpha)) - \underline{\phi}(\bar{q}(\alpha)) \geq \delta$. The minimum of the integral $\int_{[0,1]} \phi(\bar{q}(\alpha), f(\alpha)) - \underline{\phi}(\bar{q}(\alpha))d\alpha$ is attained for a trapezoidal or triangle shape, with the maximum attained at a bound (0 or 1), of length $\min(1, \frac{\delta}{8B_\Phi + \beta_\Phi \beta})$ (because the domain is of length 1). We can conclude

  $$\mathcal{R}_{\Phi, P}(f) - \inf_{g: \mathcal{X} \to \mathbb{R}^n} \mathcal{R}_{\Phi, P}(g) = \int_{[0,1]} \big(\phi(\bar{q}(\alpha), f(\alpha)) - \underline{\phi}(\bar{q}(\alpha))\big)d\alpha \geq \frac{1}{2}\min(\delta, \frac{\delta^2}{8B_\Phi + \beta_\Phi \beta})$$

- If $\{z, z'\} \not\subseteq \mathrm{pred}(f([0,1]))$:

  Assume for instance $z \notin \mathrm{pred}(f([0,1]))$. Let $\epsilon = \min_{z'' \neq z} \ell(q^{(z)}, z'') - \underline{\ell}(q^{(z)})$. For $\alpha < 1/4$, using $\underline{\ell}(\bar{q}(\alpha)) = \ell(\bar{q}(\alpha), z)$, we have:

  $$\ell(\bar{q}(\alpha), \mathrm{pred}(f(\alpha))) - \underline{\ell}(\bar{q}(\alpha)) \geq (1 - 2\alpha)\big(\ell(q^{(z)}, \mathrm{pred}(f(\alpha))) - \ell(q^{(z)}, z)\big) \geq \frac{1}{2}\epsilon.$$

  By uniform calibration (Theorem 30), there is $\delta_z'$ such that $\forall \alpha \in [0, \frac{1}{4})$, $\phi(\bar{q}(\alpha), f(\alpha)) - \underline{\phi}(\bar{q}(\alpha)) \geq \delta_z'$ (or we would have $\mathrm{pred}(f(\alpha)) = \{z\} = \mathrm{argmin}\, \ell(\bar{q}(\alpha), .)$). We then have

  $$\int_{\alpha \in [0,1]} \big(\phi(\bar{q}(\alpha), f(\alpha)) - \underline{\phi}(\bar{q}(\alpha))\big)d\alpha \geq \int_{\alpha \in [0, \frac{1}{4}]} \big(\phi(\bar{q}(\alpha), f(\alpha)) - \underline{\phi}(\bar{q}(\alpha))\big)d\alpha \geq \frac{1}{4}\delta_z'$$

  Similarly if $z' \notin \mathrm{pred}(f([0,1]))$.

Taking $c' = \min(\frac{1}{4}\delta_z', \frac{1}{4}\delta_{z'}', \frac{\delta}{2})$ and $c = \frac{\delta^2}{2}$ gives the result. $\qquad\square$

### H.4 Local valleys of calibrated surrogate losses

We now prove that when the task loss does not have connected argmins, the set of distributions in $\mathcal{Q}$ for which $\phi(q, .)$ has bad local valleys has non-zero measure.

**Proposition 51.** *Under Assumption (A5), the set $\{q \in \mathcal{Q} : \phi(q, .) \text{ has bad local valleys}\}$ has non-zero Lebesgue measure.*

*Proof.* Let $q_0$ be a distribution such that $\mathrm{argmin}\, \ell(q_0, .)$ is disconnected, and let $z \in \mathrm{argmin}\, \ell(q_0, .)$. Let $\bar{q}, \delta_0$ and $\eta_0$ be defined as in point *ii)* of Lemma 49, i.e., $\bar{q}$ is a segment between $q_0$ and $q^{(z)}$.

First let $\delta_1 \in (0, \delta_0)$, such that $\mathrm{pred}(\mathrm{lev}_{\delta_1} \phi(q_0, .)) = \mathrm{argmin}\, \ell(q_0, .)$. Such a $\delta_1$ exists using the equality of argmins for calibrated surrogate losses (Theorem 28).

Let $s' \in \mathrm{lev}_{\delta_1} \phi(q_0, .)$ such that $\mathrm{pred}(s')$ is not in the same connected component of $\mathrm{argmin}\, \ell(q_0, .)$ as $z$. Using the Lipschitz property of $q \mapsto \phi(q, s) - \underline{\phi}(q)$ (Lemma 48), we have:

$$\forall \alpha \in [0,1], \forall q \in \mathcal{B}_1(\bar{q}(\alpha), \eta), \quad \phi(q, s') - \underline{\phi}(q) < \delta_1 + (2\eta + 4\alpha)B_\Phi. \tag{9}$$

Moreover, for every $\alpha > 0$, using point *iii)* of Lemma 49 with $q := \bar{q}(\alpha)$, we can find $\delta_\alpha \in (0, \delta_1)$ and $\eta_\alpha > 0$ such that:

$$\forall s \in \mathrm{lev}_{\delta_\alpha} \phi(\bar{q}(\alpha), .), \quad \forall q \in \mathcal{B}_1(\bar{q}(\alpha), \eta_\alpha), \mathrm{pred}(s) = \{z\}.$$

Thus, for any $s' \in \mathrm{lev}_{\delta_1} \phi(q_0, .)$ such that $\mathrm{pred}(s')$ is not in the same connected component as $z$ is not in $\mathrm{lev}_{\delta_\alpha} \ell(\bar{q}(\alpha), .)$, we have

$$\forall \alpha \in [0,1], \forall q \in \mathcal{B}_1(\bar{q}(\alpha), \eta_\alpha), \quad \phi(q, s') - \underline{\phi}(q) \geq \delta_\alpha + (2\eta_\alpha + 4\alpha)B_\Phi. \tag{10}$$

Finally, since $z$ and $z'$ are not connected in $\mathrm{argmin}\, \ell(q_0, .)$, for every continuous function $f : [0,1] \to \mathbb{R}^n$ with $z \in \mathrm{pred}(f(0))$ and $z' \in \mathrm{pred}(f(1))$, we must have $\mathrm{pred}(f([0,1])) \not\subseteq \mathrm{argmin}\, \ell(q_0, .)$ (since continuous functions preserve connectedness), and thus, by our choice of $\delta_0$ from Lemma 49, we have:

$$\forall \alpha \in [0,1], \forall q \in \mathcal{B}_1(\bar{q}(\alpha), \eta_0) \max_{t \in [0,1]} \left( \phi(q, f(t)) - \underline{\phi}(q) \right) \geq \delta_0. \tag{11}$$

The proof finishes by taking $\alpha = \frac{\delta_0 - \delta_1}{12 B_\Phi}$, $\eta = \min(\eta_\alpha, \eta_0, \alpha)$. With these values, we have that $\forall q \in \mathcal{B}_1(\bar{q}(\alpha), \eta)$:

1. Using (9), there exists $s' \in \mathrm{lev}_{\delta_1 + (2\eta + 4\alpha)B_\Phi} \phi(q, .)$ such that $\mathrm{pred}(s')$ is in $\mathrm{argmin}\, \ell(q_0, .)$ but not connected to $z$ in $\mathrm{argmin}\, \ell(q_0, .)$.

2. Using $\delta_1 + (2\eta + 4\alpha)B_\Phi < \delta_0$ and (11), $s'$ above and its connected component $C$ in $\mathrm{lev}_{\delta_1 + (2\eta + 4\alpha)B_\Phi} \phi(q, .)$ are not connected to any $s$ such that $\mathrm{pred}(s) = \{z\}$. Thus, the connected component $C$ is a local valley of $\ell(q, .)$,

3. Using (10), the infimum over $C$ is suboptimal. Thus $C$ is a bad local valley.

Thus, the measure of bad local valleys is at least the measure of $\mathcal{B}_1(\bar{q}(\alpha), \eta)$, and is thus $> 0$.

$\square$