[Reviews · NeurIPS 2020]

Review 1

Summary and Contributions: This paper considers a problem of supervised ranking where the goal is to learn a mapping from (query) features to ranking over items given data about relevance of these items. In this problem there is a target loss L such as NDCG, AP etc., and the goal is to find a continuous surrogate loss such that minimizing this surrogate loss effectively corresponds to minimizing the target loss, thereby resulting in a consistent algorithm. A popular surrogate for this problem is the squared loss. Here one finds a utility for each item depending upon the underlying loss and relevance scores for the item, and tries learn a real-valued scoring function for each item by minimize the squared error between the utility and the score over the data. The ranking can then be an argsort over this scoring function. This paper studies the limits of this approach for this problem of ranking. The main result of the paper is a characterization of when such an approach based on squared loss will result in a consistent algorithm. This characterization is based upon the continuity of level sets of the loss function. The paper then elaborates more on this theory and argues that any other convex surrogate can be consistent if and only if the squared loss is consistent. Further, the paper shows that in the case squared loss is not consistent, it is possible to have non-convex consistent surrogates, however, these non-convex surrogates are guaranteed to have bad local minima.

Strengths: This paper contributes important results that shed light on the difficulty of finding surrogates for ranking loss functions. One of the main insight from the paper is that squared loss regression essentially captures the entire class of convex calibrated surrogate losses if we restrict our attention to surrogates with dimension n, and beyond this only non-convex surrogates are possible. Overall, I think this is a good set of results which are relevant to the NeurIPS community.

Weaknesses: One weakness I have some questions for the authors discussed below.

Correctness: I think the claims are sound to the best of my knowledge. However, I have not gone through the proofs in detail.

Clarity: The paper is well-written.

Relation to Prior Work: A relation to prior work is provided. A couple of references that are missing are-- 1. "Convex Calibrated Surrogates for Low-Rank Loss Matrices with Applications to Subset Ranking Losses" by Ramaswamy et al. 2. "On Consistent Surrogate Risk Minimization and Property Elicitation" by Agarwal & Agarwal

Reproducibility: Yes

Additional Feedback: 1. Squared loss surrogates for dimensions other than n are given in "Convex Calibrated Surrogates for Low-Rank Loss Matrices with Applications to Subset Ranking Losses". In the future it would be interesting to understand whether a similar result holds for these surrogates of different dimensions, i.e. does a convex calibrated surrogate in a given dimension exist if and only if there is a squared loss that is consistent for that dimension? 2. The paper "On Consistent Surrogate Risk Minimization and Property Elicitation" by Agarwal & Agarwal shows that squared losses essentially elicit a linear property of the distribution P(\cdot \vert x). Your result shows (I think) that one does not need to look beyond linear properties (also called standardization functions in Buffoni et al., 2011) of dimension n if the goal is to find a convex surrogate loss of dimension n. I think if one can find conditions on the loss matrix under which this result holds for any arbitrary dimension then using Theorem 7 in Agarwal & Agarwal, 2015 it would imply that the CCDim of the loss function is equal to the rank of the loss matrix. 3. In experiments in Figure 2, it would be interesting to understand how a noisy version of gradient descent performs. Typos: 1. Eq. after line 102: inf is over s \in \R^n. 2. Line 113: u is a function from \Y to \R^n and not \R


Review 2

Summary and Contributions: The authors elaborate on the connection between ranking and regression. More specifically, they consider the question which type of ranking problems (i.e., loss functions for ranking) can be solved by means of standard regression (minimizing L2 loss), in the sense that the latter is asymptotically consistent. As a main result, they provide a corresponding characterization of ranking losses, from which they derive two important implications. First, if a consistent approach based on convex risk minimization exists for the ranking problem, then there is also a consistent approach based on regression. Second, in cases where regression is not consistent, there are data distributions for which consistent surrogate approaches (scoring functions) necessarily have non-trivial local minima.

Strengths: -- relevant and timely topic -- strong theoretical result

Weaknesses: -- clarity and accessibility could still be improved

Correctness: Seems correct. However, the paper comes with an appendix of more than 25 pages of proofs etc. It's impossible to check everything carefully within the short period for reviewing.

Clarity: Mostly yes.

Relation to Prior Work: Yes.

Reproducibility: Yes

Additional Feedback: Thanks for the rebuttal. Original review: Overall, I think this is a really nice paper that makes a significant contribution. The results are non-trivial and provide a deep insight into the connection between ranking and regression. The paper is well presented and appears to be technically sound, although I didn't check all proofs in detail. I only have a few minor remarks: What I found confusing is the use of the "supervision" Y. What exactly is meant by this? In the first paragraph in Section 2.1, \mathcal{Y} is said to be the space of "supervision signals", and the loss L takes a tuple (Y,pi) as input, where pi is a predicted ranking. Normally, a loss compares a prediction with the corresponding ground truth, but it seems a supervision signal is not a ground truth ranking. Later, the (latent) utility function u is also defined on \mathcal{Y}, so here its seems to be a set of items to be scored. "In recommender systems or search engines, this means that the score of an item depends on the other available items" --> is this consistent with defining a utility function on individual items? If items are scored in this way, then their score is independent of other items available. "In multilabel classification, in some rare cases, efficient inference procedures have been found [31], but not yet in ranking" --> Perhaps the following papers could be relevant: Kotlowski et al., Bipartite Ranking through Minimization of Univariate Loss, ICML 2011. Kotlowski et al.,Consistent Multilabel Ranking through Univariate Loss Minimization, ICML 2012.


Review 3

Summary and Contributions: Ranking is a combinatorial problem, but if it is formidable as a regression + sorting, then it becomes much easier and faster to implement. This paper presents exact ("if and only if") characterization of when this is indeed the case, in terms of the underlying utility function.

Strengths: Looks very solid, with fundamental results. However it is dense and builds up on a lot of prior work. Even after two passes I did not digest all aspects of the work.

Weaknesses: A minor rewrite could make it accessible to a wider audience. For example, they could provide a table to give examples of relatable expected utilities will meet the criterion (rather than just say "sublevel sets are connected" + Th 10), and some examples of when it is not.

Correctness: methods is likely correct. This is a theory paper, so not much empirical here.

Clarity: yes, but dense.

Relation to Prior Work: yes

Reproducibility: Yes

Additional Feedback: UPDATE: I have read and taken into account the rebuttal, as well as any ensuing discussions.


Review 4

Summary and Contributions: This paper answer the question of when is square loss regression consistent for ranking via score-and-sort by finding a characterization of ranking losses, which could also be used to obtain two important collaries. These results not only provide a good understanding of different surrogate approaches for ranking, and also illustrate the intrinsic difficulty of solving general ranking problems with the score-and-sort approach.

Strengths: 1. The addressed problem is well motivated. 2. The propose results are theoretically solid. 3. The necessary non-global minima of surrogate losses and the discontinuity of optimal scoring are novel.

Weaknesses: 1. Lack empirical validations to support the theoretical findings.

Correctness: The theoretical study of using calibration, connectedness, discontinuity of optimal scoring functions, and bad local minima is technically sound. Though I did not have time to check the correctness of each theorem and their proof in detail, the results are intrinsically understandable.

Clarity: The paper is well written. The motivation is clearly illustrated, and their contribution is well concluded.

Relation to Prior Work: Discussions of this work with most previous related works have been conducted. Two more papers on ranking consistency should be added to the related work: Statistical Consistency of Top-k Ranking, NIPS 2009. Statistical Consistency of Ranking Methods in A Rank-Differentiable Probability Space. NIPS 2012.

Reproducibility: Yes

Additional Feedback:

[Author Response · NeurIPS 2020]

We first would like to thank all reviewers for their reviews and constructive comments. We updated the paper to take into account the suggestions and corrections that were proposed: we

- (obviously) corrected typos and other minor formulation errors
- expanded on the related work and discussions, including adding references suggested by R1, R2, R4
- clarified section 2.1 following our answer to R2's questions
- added graphical representations of connected/disconnected sublevel sets to complement Figure 1, with a table of formula of different ranking losses and associated utilities if any, following R3's suggestion

We give more details on some discussion points below.

**R1: "2. [...] it would imply that the CCDim of the loss function is equal to the rank of the loss matrix."** Yes. In fact, the symmetry assumption in our definition of a ranking loss (" Items are equivalent a priori" in Definition 3) implies that ranking losses satisfy the assumptions of Theorem 18 in Ramaswamy & Agarwal [22]. So for a convex calibrated loss, we do have CCdim $\geq$ affdim(L) - 1, which is what Theorem 7 of Agarwal & Agarwal would give. Note that these two theorems use a definition of "calibration" that does not use argsort as the inference procedure. In their case, inference may be intractable. Thanks for this remark, we will add it.

**R1: "does a convex calibrated surrogate in a given dimension exist if and only if there is a squared loss that is consistent for that dimension?"** Indeed, it would be interesting to extend our analysis to higher dimensions, for fixed "interesting" inference procedures other than argsort. Note that we need to focus on fixed inference schemes: If we accept possibly intractable inference procedures, the approach of Ramaswamy & Agarwal (2012), based on decomposing the loss matrix, works to define calibrated square losses in any dimension.

**R2: "the loss L takes a tuple (Y,pi) as input, where pi is a predicted ranking. Normally, a loss compares a prediction with the corresponding ground truth, but it seems a supervision signal is not a ground truth ranking."** We agree with the reviewer that in most supervised learning tasks, the supervision is a ground truth in prediction space (a ranking in our case). Yet, in many practical ranking tasks, the supervision is not a complete ranking. For instance, in search engines, the task is to rank documents in response for a query. A typical setup is when annotators give binary relevance judgments to each document given a query. The set of relevance judgments does not define a full ranking, because it does not specify the relative order of two documents with the same relevance. By decoupling the supervision space $\mathcal{Y}$ from the prediction space $\mathfrak{S}_n$, our framework is more general than a standard supervised learning framework since it allows for $\mathcal{Y} = \mathfrak{S}_n$, but also for other supervisions such as relevance judgements.

**R2: ""In recommender systems or search engines, this means that the score of an item depends on the other available items" –> is this consistent with defining a utility function on individual items?"** For a utility function, we can say *the input of the utility function is the entire supervision (e.g., all relevance judgments for all items to rank), and it computes jointly the utility values for all items*. The sentence quoted by the reviewer makes the analogous statement for scoring functions: *the input of the scoring function are the features of all items to rank, and it jointly computes the scores of all items*. These are consistent: there is a utility value per item on one side, and one score per item on the other side.

[Meta-Review · NeurIPS 2020]

All of the reviewers are quite positive about the paper and unanimously agree on acceptance.